# Traditional and Complementary Medicine Use among Cancer Patients in Asian Countries: A Systematic Review and Meta-Analysis

**DOI:** 10.3390/cancers16183130

**Published:** 2024-09-11

**Authors:** Soojeung Choi, Sangita Karki Kunwor, Hyeabin Im, Dain Choi, Junghye Hwang, Mansoor Ahmed, Dongwoon Han

**Affiliations:** 1Department of Global Health and Development, Graduate School, Hanyang University, Seoul 04763, Republic of Korea; soojeungchoi@gmail.com (S.C.); hbim191@gmail.com (H.I.);; 2Institute of Health Services Management, Hanyang University, Seoul 04763, Republic of Korea; 3Department of Preventive Medicine, College of Medicine, Hanyang University, Seoul 04763, Republic of Korea; 4Department of Obstetrics and Gynecology, College of Medicine, Hanyang University, Seoul 04763, Republic of Korea; 5School of Public Health, Dow University of Health Sciences, Karachi 75330, Pakistan

**Keywords:** traditional and complementary medicine, integrative oncology, Asia

## Abstract

**Simple Summary:**

The demand for traditional and complementary medicine (T&CM), perceived as natural and safe, among cancer patients has been steadily increasing. However, this trend raises concerns about the potential risks of using these therapies concurrently with conventional treatments and whether patients are fully adhering to their prescribed treatments. In Asia, T&CM is particularly common due to cultural and historical influences. This systematic review and meta-analysis is the first study to investigate the prevalence of T&CM use among cancer patients in Asia, how often they disclose this use to their physicians, and the factors influencing their choices. Understanding these aspects may enhance communication between patients and physicians, ultimately leading to safer and more effective cancer care.

**Abstract:**

Globally, cancer patients frequently use T&CM during their treatment for various reasons. The primary concerns regarding the use of T&CM among cancer patients are the potential risks associated with interactions between pharmaceuticals and T&CM, as well as the risk of noncompliance with conventional cancer treatments. Despite the higher prevalence of T&CM use in Asia, driven by cultural, historical, and resource-related factors, no prior review has tried to estimate the prevalence and influencing factors of T&CM use and disclosure among cancer patients in this region. This study aims to examine the prevalence and disclosure rates of T&CM use among cancer patients in Asia to assess various factors influencing its use across different cancer treatment settings in Asia. Systematic research on T&CM use was conducted using four databases (PubMed, EMBASE, Web of Science, and CINAHAL) from inception to January 2023. Quality was assessed using the Appraisal Tool for Cross-Sectional Studies (AXIS). A random effects model was used to estimate the pooled prevalence of T&CM use, and data analysis was performed using Stata Version 16.0. Among the 4849 records retrieved, 41 eligible studies conducted in 14 Asian countries were included, involving a total of 14,976 participants. The pooled prevalence of T&CM use was 49.3%, ranging from 24.0% to 94.8%, and the disclosure rate of T&CM use was 38.2% (11.9% to 82.5%). The most commonly used T&CM modalities were herbal medicines and traditional medicine. Females were 22.0% more likely to use T&CM than males. A subgroup analysis revealed the highest prevalence of T&CM use was found in studies conducted in East Asia (62.4%) and those covered by both national and private insurance (55.8%). The disclosure rate of T&CM use to physicians remains low. Moreover, the factors influencing this disclosure are still insufficiently explored. Since the disclosure of T&CM use is a crucial indicator of patient safety and the quality of cancer treatment prognosis, future research should focus on identifying the determinants of non-disclosure.

## 1. Introduction

The increasing burden of cancer is a pressing global healthcare concern. In 2022, an estimated 9.7 million individuals worldwide succumbed to cancer, with approximately 56.1% of these deaths occurring in Asia [1]. The Asian region, which is predominantly composed of developing and underdeveloped countries, struggles with growing cancer challenges attributed to its weak healthcare system and various sociodemographic or environmental factors [2,3]. Despite notable advances and refinements in cancer treatment [4,5,6], many patients in Asia seek traditional and complementary medicine (T&CM) as a culturally accessible and readily available strategy to address their unmet healthcare needs [7]. T&CM combines the terms traditional medicine (TM) and complementary medicine (CM), encompassing a range of products, practices, and practitioners. TM has a history that spans thousands of years and is growing worldwide. CM refers to a wide range of healthcare practices outside the dominant healthcare system, often used interchangeably with traditional medicine in some countries [8].

The advantages of T&CM in health management, well-being, and patient-centered care are leading to its progressive integration into healthcare systems globally [9,10]. Integrating T&CM into cancer care has shown promise in reducing the discomfort associated with cancer therapy [11] and enhancing disease control rates [12]. Curcumin, commonly known as turmeric, exhibits anticancer activity across various cancer types [13]; ginger effectively alleviates chemotherapy-related nausea and vomiting [14]; and St. John’s wort is efficacious in managing depression and anxiety [15,16]. However, the use of T&CM requires caution due to potential drug interactions, which may attenuate therapeutic effects or exacerbate adverse drug reactions [16,17,18]. Additionally, awareness of the potential risks associated with T&CM is crucial. Studies have shown that patients who choose to use complementary therapies are 41% more likely to experience breast cancer recurrence [19] or refuse conventional cancer treatment, which can significantly affect their chances of survival [20]. Furthermore, the use of T&CM may impact a patient’s adherence to the chemotherapy regimen prescribed by oncologists [21]. Therefore, gaining a comprehensive understanding of T&CM use among cancer patients has important implications for health professionals and health policymakers involved in cancer care.

The prevalence and patterns of T&CM use among adult cancer patients vary considerably between studies. Previous systematic reviews have examined the use of T&CM in this population, with overall prevalence estimates ranging from 31.4% to 54.0% [22,23,24,25,26,27,28,29]. While certain research encompasses all cancer types, other researchers focus on specific cancers, such as breast cancer [30] and gynecological cancers [26]. Some studies have examined the use of T&CM among cancer patients in particular regions, including the British Commonwealth of Nations [27], in low-income and lower-middle-income countries [29], and in sub-Saharan Africa [31]. From a methodological point of view, only two studies conducted a meta-analysis [24,32]. Non-biomedical approaches in Asia have been deeply intertwined with various historical, religious, and cultural contexts, evolving in tandem with their own cultures. Access to and consumption of T&CMs persist among significant portions of the Asian population, despite a substantial reliance on biomedical treatments. However, no attempts have been made to consolidate studies to establish a robust prevalence estimate of T&CM use among Asian cancer patients or analyze the impact of sociodemographic factors on its utilization. In addition, the factors influencing disclosure of T&CM use to the physician, which may affect adherence to conventional care or potential hazards from drug-T&CM interactions, remain poorly understood [23,33].

The objectives of this review were to systematically review and synthesize the evidence on the prevalence of T&CM use and disclosure among adult cancer patients in the Asian region and identify factors influencing variations in prevalence estimates of T&CM, such as gender, cancer type, geographical region, health insurance coverage, income level, and survey methods, and to determine if the use of T&CM is consistent across Asia.

## 2. Materials and Methods

The study protocol was registered on PROSPERO (CRD42021252288), and the methods and results of this review were reported according to the Preferred Reporting Items for Systematic Reviews (PRISMA) guidelines [34].

### 2.1. Eligibility Criteria

Studies that reported the prevalence of T&CM use by cancer patients were eligible for inclusion in the review. To address variations in population, which is a recognized source of heterogeneity, the sampling frame was limited to cancer patients currently receiving treatment at cancer centers or hospitals. Studies involving patients attending follow-up appointments after finishing their cancer treatments were not included. Furthermore, only studies that clearly indicated the use of T&CM as either “yes” or “no” were considered for the analysis.

Inclusion criteria were organized based on the CoCoPop (condition, context, and population) framework developed for systematic reviews of prevalence and incidence [35]. The criteria were as follows: (i) condition: any type of cancer patients; (ii) context: the use of T&CM by cancer patients in Asian countries; (iii) population: cancer patients aged 18 years and older who are undergoing active cancer treatments; (iv) study design: observational studies with primary research findings. Exclusion criteria included the following: (i) studies conducted in facilities that provided both conventional medicine and T&CM, or that were unable to determine the prevalence of T&CM; (ii) cancer patients who had completed all cancer treatments or were in the palliative care phase; (iii) pediatric patients or participants who were not patients; (iv) publications identified as clinical trials (RCTs, animal trials, etc.), reviews, dissertations, pilot studies, editorials, commentaries, and evaluations of effectiveness on a single modality.

### 2.2. Information Sources and Search Strategy

The inclusion and exclusion criteria listed in the research protocol formed the basis for the construction of search strategies. PubMed, EMBASE, Web of Science, and CINAHL were systematically searched to identify eligible articles published from inception to January 2023 using the MeSH/Emtree terms “cancer/neoplasm”, “complementary therapies”, and “traditional medicine”. Additionally, the reference lists of included studies were manually searched in Google Scholar to identify additional studies. The list of Asian countries was compiled from the classification criteria from the United Nations report [36]. The entire search string for each database is available in the Appendix A (see online Appendix A).

### 2.3. Selection Process

The citations retrieved from the databases were extracted into Endnote 20 (Clarivate Analytics) to eliminate duplicates. Three authors (SC, HI, and DC) independently screened the titles and abstracts of each study using the pre-defined inclusion criteria and categorized the publications into “not relevant” or “potentially relevant”. Disagreements were resolved by unanimity through discussion.

### 2.4. Data Collection Process

Data extracted from the included records were arranged in a predefined Excel spreadsheet to facilitate data extraction. Two authors (SC and SK) independently conducted the data extraction. In cases where the findings lacked sufficient statistical information or clarity, the primary author (SC) contacted the original authors to obtain additional details, successfully obtaining detailed information from them. Studies that did not provide complete quantitative details were excluded from our data analyses.

### 2.5. Data Items

Data items were extracted across three categories: (a) study characteristics, including author, year of publication, country, aim of the study, study design and setting, sample size, and number of participants; (b) population demographics and clinical characteristics of participants; and (c) main findings related to the concurrent use of cancer treatment and T&CM, such as the prevalence of T&CM use, types of T&CM modalities, disclosure rates of T&CM use to healthcare providers, reasons for non-disclosure, and physicians’ responses when the use of T&CM was disclosed.

### 2.6. Study Risk of Bias Assessment

The methodological quality of the included studies was assessed using the “Appraisal Tool for Cross-Sectional Studies (AXIS)” [37]. The AXIS, which was developed in 2016 and is the recently preferred one, contains 20 items categorized into three domains: seven items (1, 4, 10, 11, 12, 16, and 18) for the quality of reporting, seven for the appropriateness of study design (2, 3, 5, 8, 17, 19, and 20), and six items (6, 7, 9, 13, 14, and 15) for the risk of bias in the study. As this tool does not provide a numerical scale for assessment, we computed the total number of “yes” responses of each item, with each paper assigned a score out of 20 based on its fulfillment of the specified criteria [38,39]. However, in our study, Question 14 is considered an extension of Question 7, and no points are allocated for Question 14. Consequently, the maximum attainable score is set at 19 points. The reviewers (SC, SK, HI, and DC) reached a consensus on the scoring through discussions and consultations with the third reviewers (MA, JH, and DW). The overall scores were classified as follows: a low risk of bias (15–19) for the top third, a moderate risk of bias (10–14) for the second third, and studies scoring ten or below were considered to be at a high risk of bias [40].

### 2.7. Synthesis Methods and Reporting Bias Assessment

The primary outcome was the pooled prevalence estimate of T&CM use, with a confidence interval of 95% (95% CI). Data analysis was conducted using the Metaprop command, an extension of the metan command in Stata version 16.1 (Stata Corp LP, College Station, TX, USA), specifically developed for meta-analysis of proportions [41]. The use of the Metaprop command allows for the inclusion of studies with proportions of 0% or 100% in the meta-analysis, while also preventing confidence intervals that exceed the range from 0 to 1 [41]. The pooled prevalence estimate of T&CM use was calculated using the Freeman–Tukey double arcsine transformation to stabilize the variances. The corresponding 95% CIs were calculated using the Score (Wilson) exact method. A random effects model was used for data pooling, and the results were visually presented using forest plots. Cochran’s Q tests and I-squared statistics (I^2^) were used to evaluate heterogeneity. Heterogeneity was quantified using the Cochran’s Q test with a significance level of *p* < 0.1. The I^2^ statistic was used to quantify the extent to which the observed variation across studies can be attributed to heterogeneity rather than chance [42].

To investigate the potential sources of heterogeneity in the meta-analyses, subgroup analyses were conducted based on year of survey, sample size calculation, data collection methods, geographic locations, income level economics, gender, health insurance, and types of cancer. The survey years were categorized according to the year of publication of the WHO Report on Traditional Medicine Strategy 2014–2023, which includes the updated strategy emphasizing health services and systems, including T&CM products, practices, and practitioners [9]. Given Asia’s extensive diversity, including variations in cultural, economic, and healthcare contexts, the region was classified into four distinct geographic areas to evaluate potential differences in T&CM use: Western Asia (Iran, Lebanon, Turkey), South Asia (India, Nepal, Pakistan), Southeast Asia (Indonesia, Malaysia, Singapore, Thailand), and East Asia (Korea, China, Mongolia, Taiwan). Income level classifications (2021–2022) were obtained from the World Bank: high-income economies (>USD 12,695), upper-middle-income economies (USD 4096–USD 12,695), and lower-middle-income economies (USD 1046–USD 4095). Health insurance coverage for T&CM was categorized based on World Health Organization data as follows [8]: (1) national health insurance (Iran, Lebanon, Thailand, Mongolia), (2) private health insurance (Turkey, Malaysia, Singapore), (3) both (China, India, Korea, and Taiwan), and (4) T&CMs are not covered by health insurance in Indonesia, Nepal, and Pakistan. Review manager software version 5.4 was also used to compare the odds ratio of T&CM utilization between males and females. Funnel plots and Egger’s test were used to examine the potential publication bias [43,44].

## 3. Results

### 3.1. Study Selection

A flowchart illustrating the search process is presented in the PRISMA flow diagram (Figure 1). The initial search of databases retrieved 4849 citations, and an additional 20 records were identified from the reference lists of included studies. After removing duplicates, 3827 records were screened for inclusion based on title and abstract. Subsequently, 189 studies were selected for full-text screening. Among the potentially eligible full-text articles, 147 studies were excluded, with the most common reason being the ineligibility of study participants (n = 69). Forty-three reports were initially selected for inclusion; however, two reports with overlapping samples were excluded. Finally, forty-one studies were included in this review (Table 1).

### 3.2. Study Characteristics

The majority of the studies were designed as cross-sectional studies in 14 Asian countries, including Turkey (n = 12) [54,65,67,71,75,76,77,80,81,83,85], Malaysia (n = 5) [50,52,57,59,61], Korea (n = 4) [47,60,78,79], India (n = 3) [48,49,51], Iran (n = 3) [45,53,56], Thailand (n = 3) [55,70,74], China (n = 2) [58,69], Lebanon (n = 2) [62,66], Singapore (n = 2) [72,73], Indonesia (n = 1) [64], Mongolia (n = 1) [63], Nepal (n = 1) [46], Pakistan (n = 1) [84], and Taiwan (n = 1) [68]. Two-thirds of the studies were published from 2011 onwards (n = 27, 65.9%). The total number of participants was 14,976, with sample sizes ranging from 100 to 2614 participants (mean (SD) = 365.3 (395.3)). The majority of participants were female (63.0%). Geographically, the studies were primarily conducted in Western Asia (n = 17, 41.5%), followed by Southeast Asia (n = 11, 26.8%). Thirty studies investigated multiple cancer types, while eleven articles focused on specific types, such as breast cancer [57,59,61,64,66,75], gynecological cancer [74], gastrointestinal cancer [82], and lung cancer [47,62]. After excluding studies that did not report individual cancer types, the most prevalent types of cancer were breast cancer (32.7%), followed by gastrointestinal cancer (14.5%), lung cancer (10.2%), and hematologic cancer (including leukemia, lymphoma, and multiple myeloma) (8.1%).

### 3.3. Risk of Bias in Studies

Twenty-two (53.7%) studies demonstrated a substantial fulfillment of the criteria according to the AXIS tool, indicating a high methodological quality and low susceptibility to bias. The quality assessment scores for each domain were 6.54 out of 7.0 points for study design, 5.90 out of 7.0 points for reporting quality, and 2.83 out of 5.0 points for risk of bias. All studies clearly stated their aims and target populations and employed appropriate study designs. However, areas of improvement were identified, such as justifying the sample size, addressing and categorizing non-responders, and providing information on the reliability of measurements. The detailed, individual risk-of-bias analysis is provided in the Appendix A (see online Appendix A).

### 3.4. Results of Syntheses

#### 3.4.1. The Pooled Prevalence of T&CM Use

The pooled estimate of T&CM use was 49.3% (95% CI: 44.5–54.0% I^2^ = 97.0%, *p* < 0.000). The highest prevalence of T&CM use was observed in the study of Bazrafshani in Iran (84.8%, 95% CI: 80.4–86.3%) [53], while Dişsiz and Yilmaz’s study in Turkey reported the lowest utilization of T&CM (24.0%, 95% CI: 19.1–29.7%) [65] (Figure 2). Concerning country-level data, the highest prevalence of T&CM utilization was observed among cancer patients in China (78.5%, 95% CI: 75.2–81.5%) [58,69] and Taiwan (79.3%, 95% CI: 73.2–84.6%) [68], whereas the lowest rate was observed in Pakistan (29.0%, 95% CI: 21.6–37.3%) [84] (Figure 3). Significant heterogeneity was observed across the included studies (*p* < 0.0001; I^2^ = 97.0%). The funnel plot showed a symmetrical distribution of estimates reported in the studies (see online Appendix A).

#### 3.4.2. Subgroup Analyses

As heterogeneity was detected, we performed a subgroup analysis to explore potential sources of heterogeneity (Table 2). The prevalence estimate of T&CM use exhibited variation across different regions (*p* = 0.001), health insurance (*p* < 0.000), and survey year (*p* = 0.040). In terms of regional T&CM use, the highest pooled prevalence estimate was observed in East Asia (62.4%, 95% CI: 50.5–73.7%; I^2^ = 97.6%) [47,58,60,63,68,69,78,79], while the lowest estimate was reported in South Asia (33.5%, 95% CI: 25.1–42.6%; I^2^ = 96.0%) [46,48,49,51,84]. The heterogeneity among these subgroups was found to be significant according to the Q-test (Χ^2^ = 17.714, *p* = 0.001), indicating a significant difference in the pooled estimates of subgroups by geographical regions. The examination of the variation in utilization rates based on health insurance coverage for T&CM showed that the highest utilization rate was 55.8% (95% CI: 44.7–66.6%) when both public and private insurance covered T&CM, and the lowest was 31.7% (95% CI: 29.3–34.2%) when T&CM was not covered by insurance. A significant heterogeneity was observed among the studies based on the health insurance coverage (Χ^2^ = 42.242, *p <* 0.000).

In a sub-analysis considering the survey year, we examined the prevalence of T&CM use before 2013 [57,60,61,67,68,69,70,71,72,73,74,75,76,77,78,79,80,81,82,83,84,85] and after 2014 [45,46,47,48,49,50,51,52,53,54,55,56,58,59,62,63,64,65,66]. The prevalence was found to be 53.7% (95% CI: 47.9–59.5; I^2^ = 95.5%, *p <* 0.0001) before 2013 and 44.2% (95% CI: 37.4–51.2%; I^2^ = 97.4%, *p <* 0.0001) after 2014. Statistical significance was observed in heterogeneity when categorized by the survey year (Χ^2^ = 4.200, *p =* 0.040).

The subgroup analysis, based on whether a sample size calculation was provided, the data collection method, and income level revealed that the highest utilization of T&CM was observed among groups where a sample size calculation was not provided (49.6%, 95% CI: 44.3–54.9%) [45,46,51,57,62,66,74,78], those using self-completed questionnaires (53.6%, 95% CI: 38.0–68.9%) [47,48,52,56,58,69,71,76], and high-income countries (53.3%, 95% CI: 44.5–62.0%) [47,60,72,73,78,79]. However, there were no significant differences in the prevalence of T&CM use based on these variables.

##### Gender Differences in Pooled Estimates of T&CM Use

We examined whether the prevalence of T&CM use varies by gender. Five studies did not report the prevalence of T&CM use by gender [45,49,56,65,80]; so, thirty-six studies were included in this analysis. The pooled prevalence estimate in the combined sample was 50.2% for males (95% CI: 44.3–56.0%) and 52.3% for females (95% CI: 46.3–58.2%) (Table 2). The difference in utilization of T&CM between males and females was analyzed using 27 comparable datasets [46,47,48,50,51,52,53,54,55,58,60,62,63,67,68,69,70,71,72,73,76,77,78,79,81,83,85]. Females were 22% more likely to use T&CM than males (OR = 1.22, 95% CI: 1.07–1.39%, *p* = 0.002; I^2^ = 50%) (Figure 4).

The results of the subgroup analysis on gender differences in the use of T&CM reveal no significant heterogeneity across East Asia, high-income countries, and countries with both national and private insurance coverage for T&CM. In East Asia, females are 1.42 times more likely to use T&CM than males (95% CI: 1.20–1.68%, *p* = 0.49; I^2^ = 0%), while in South Asia, females are less likely to use it than males (OR = 0.89, 95% CI: 0.70–1.12%, *p* = 0.57; I^2^ = 0%). In countries classified as high-income and in those where both national and private insurance cover T&CM, female patients were 1.16 times (95% CI: 0.97–1.39%, *p* = 0.90; I^2^ = 0%) more likely to utilize T&CM than males and 1.27 times (95% CI: 1.07–1.50%, *p* = 0.67; I^2^ = 0%) more likely to use T&CM than males, respectively (see online Appendix A).

##### Cancer-Wise Pooled Prevalence Estimates of T&CM Use by Region

The pooled prevalence estimates of T&CM use for different cancer types based on the regions are presented in Table 3 and Appendix A. In East Asia, the highest prevalence estimates of T&CM use were observed for all cancers, except genitourinary cancer [46,52,54,58,60,63,67,68,70,71,76,77,81,83] and gastrointestinal cancer [46,52,53,54,58,60,63,67,68,69,70,71,76,77,78,79,81,83]. Genitourinary cancer had the highest pooled prevalence estimate of T&CM use in South Asia, at 77.8% (95% CI: 66.9–85.8%) [46], followed by Southeast Asia at 75.5% (95% CI: 57.6–90.7%) [52,70]. Significant heterogeneity was observed across the fourteen studies that reporting genitourinary cancer based on regions (*p* < 0.000; I^2^ = 76.2%). For gastrointestinal cancer, a minimal difference was found in the prevalence between the two regions with the highest rates. South Asia showed a prevalence of 61.8% (95% CI: 49.5–72.5%) [58,60,63,68,69,78,79], followed closely by East Asia with a prevalence of 61.2% (95% CI: 51.8–70.2%) [58,60,63,68,69,78,79]. No significant heterogeneity was seen across the studies (*p* = 0.209, I^2^ = 87.6%).

In the case of breast cancer patients, twenty-nine studies provided data on the prevalence of T&CM use [46,48,50,52,53,54,55,57,58,59,60,61,63,64,66,67,68,69,70,71,72,73,75,76,77,79,81,83,84]. The highest estimate was observed in East Asia (75.0%, 95% CI: 60.3–87.3%) [58,60,63,68,69,79], while the lowest estimate was reported in South Asia (40.6%, 95% CI: 10.8–74.9%) [46,48,84]. In East Asia, the prevalence of T&CM use among gynecological cancer patients was 71.2% (95% CI: 57.3–87.3%) [58,60,79], followed by Western Asia at 59.6% (95% CI: 33.4–83.3%) [53,76,81,82]. Significant heterogeneity was observed across the studies reporting breast cancer (*p* < 0.000, I^2^ = 76.2%) and gynecological cancer (*p* < 0.000, I^2^ = 90.4%) [48,50,52,53,55,58,60,74,76,79,81,82] by region. Twelve studies reported the prevalence of T&CM use among patients with hematologic cancer [46,52,53,54,58,67,69,71,72,76,77,81], with the highest pooled estimate observed in East Asia at 91.1% (95% CI: 76.3–99.7%) [58,69], followed by South Asia at 86.7% (95% CI: 80.7–91.1%) [46]. Significant heterogeneity was observed across the studies reporting hematologic cancer based on the regions (*p* < 0.000; I^2^ = 92.9%).

##### Cancer-Wise Pooled Prevalence Estimates of T&CM Use by Level of Economic Income 

Patients with hematologic cancer (85.8%, 95% CI: 80.8–90.1%) [46,53] and head and neck cancer (61.4%, 95% CI: 54.4–68.1%) [46,48] had the highest prevalence of T&CM use in low-middle-income countries. Conversely, high-income countries showed the highest pooled prevalence estimates of T&CM use for genitourinary cancer (66.7%, 95% CI: 39.1–86.2%) [60], gynecological cancer (65.3%, 95% CI: 56.3–73.7) [60,79], gastrointestinal cancer (64.6%, 95% CI: 58.1–70.9) [60,78,79], and breast cancer (58.8%, 95% CI: 53.7–63.9%) [60,72,78,79]. The prevalence was highest at 54.9% (95% CI: 45.3–64.3%) among lung cancer patients [52,54,58,62,67,68,69,70,71,76,77,81,83] in upper-middle-income economies, with heterogeneity statistics (Χ^2^ = 1.074, *p* = 0.585; I^2^ = 81.9%). Significant heterogeneity was observed in studies reporting the prevalence of T&CM use in patients with hematologic cancer (Χ^2^ = 44.199, *p* < 0.001; I^2^ = 92.9%) [46,52,53,54,58,67,69,71,72,76,81]. However, no significant heterogeneity was seen across the studies of other cancers (Table 3 and Appendix A).

#### 3.4.3. T&CM Modality

Out of the 41 studies reviewed, T&CM modalities were categorized according to the NCCIH classification method, except for 4 studies [45,64,78,82] that did not include specific details or quantities of modalities (see online Appendix A). Among Asian cancer patients, the most commonly used nutritional modalities were herbal medicines (n = 1480), dietary supplements (n = 815), vitamins/minerals (n = 718), stinging nettle (n = 625), and ginger (n = 356). The frequently utilized psychological and physical approaches included yoga (n = 492), praying (n = 465), meditation (n = 241), and massage (n = 200). Additionally, traditional medicine was the most frequently used other complementary health approach (n = 1005). However, when specific herbs, such as stinging nettle, are incorporated into herbal medicines, they demonstrate the highest prevalence among herbal medicine users.

#### 3.4.4. Predictors of T&CM Use

Nineteen studies (46.3%) investigated the potential predictors affecting T&CM use during cancer treatments. Two of these studies did not find statistically significant variables [47,66], while the remaining seventeen identified predictive factors [46,48,52,53,55,57,58,59,61,62,63,64,65,72,78,81,84]. The most frequently reported predictive factors for T&CM use were being female [63,78,81], higher household income [48,78,81], and experiencing symptoms [53,60,68]. Education level and cancer stage yielded contrasting results. Two studies found that higher education was a significant predictor of T&CM use [63,72], while another study indicated that a lower education level was a potential factor in T&CM use [64]. Similarly, in relation to the cancer stage, two studies reported that the use of T&CM use was more common among patients in advanced stages [81,84]. However, one study found a higher T&CM use among patients in the early stages [46]. Employment status was identified as an influencing factor in T&CM use in the Lebanon study [62], while unemployment was explored as a predictive factor in India [48]. Other predictive factors for T&CM use included previous positive experience with T&CM use [63,72], prior knowledge about T&CM [65], ethnicity [62,72], younger age [63], urban residence [53], change in outlook on life after the development of cancer [52], use of multiple chemotherapy applications [81], and lower trust in physicians [64] (see online Appendix A).

#### 3.4.5. Disclosure Rate of T&CM Use

A total of twenty-six studies [46,48,50,52,53,54,58,59,60,62,63,65,66,67,68,69,70,72,73,76,77,79,81,83] reported the disclosure rate of T&CM use among cancer patients (Table 2, Figure 5). The pooled prevalence estimate of disclosing T&CM use among cancer patients was 38.2% (95% CI: 30.4–46.3%), ranging from 11.9% [67] to 82.5% [50], with a large amount of heterogeneity (Χ^2^ = 703.5, *p* < 0.001; I^2^ = 96.4%). When examining different regions, the highest estimate of disclosing T&CM use was observed in the Southeast Asian region (51.1%, 95% CI: 34.6–67.4%), while the lowest estimate was found in the Western Asian region (29.0%, 95% CI: 20.1–38.7%). The pooled prevalence estimates of disclosing T&CM use were higher in studies conducted since 2014 (42.9%, 95% CI: 30.7–55.6%), with sample sizes below 300 (43.2%, 95% CI: 33.2–53.5%), and in high-income countries (43.5%, 95% CI: 33.0–54.3%). However, no significant heterogeneity was observed in the studies included in the subgroup analyses.

#### 3.4.6. Reasons for Non-Disclosure and Physicians’ Reactions to Disclosure of T&CM Use

The reasons for the non-disclosure of T&CM use among cancer patients were examined across fourteen studies (Table 4) [46,48,50,52,58,59,62,63,68,69,70,73,79,81]. The most frequently reported reason for non-disclosure was “the lack of inquiry by the doctor regarding T&CM use”, identified in nine studies [46,48,52,59,63,68,69,70,73]. This emerged as the primary reason for non-disclosure in six of those studies [46,48,52,63,68,69]. Additionally, seven studies [46,50,62,63,68,70,79] indicated patients’ perceived lack of necessity to discuss their T&CM use with physicians, which was identified as the primary reason for non-disclosure in four studies [50,62,70,79].

Among the twenty-six studies examining the disclosure of T&CM use, seven studies investigated the physicians’ reactions to the patients who disclosed their T&CM use [50,52,62,66,70,73,81]. The pooled estimate of physicians’ positive response to the use of T&CM was 36.5% (95% CI: 21.7–52.5; X^2^ = 67.3, *p* < 0.001; I^2^ = 91.1%), ranging from 20.0% [66] to 66.4% [73]. Conversely, the pooled estimate of negative responses was lower, at 26.0% (95% CI: 7.9–49.6%; X^2^ = 156.5, *p* < 0.001; I^2^ = 96.2%). The studies reported a wide range in the percentage of patients perceiving unfavorable responses from physicians regarding the use of T&CM, ranging from 4.9% [81] to 65.0% [50]. The pooled estimate of the neutral response was 32.2% (95% CI: 18.2–47.9%; X^2^ = 66.0, *p* < 0.001; I^2^ = 90.9%), ranging from 13.9% [50] to 70.5% [54].

## 4. Discussion

This comprehensive review examines the use of T&CM among cancer patients in Asia and explores the disclosure of their T&CM use to their physicians. We compiled data from forty-one studies from 14 Asian countries. Approximately 50% (ranging from 24.0% to 94.8%) of cancer patients in Asia reported utilizing T&CM, which is comparable to recent systematic reviews by Keene et al., who reported a global prevalence of 51.0% [28], and Hill et al., who found a prevalence of 54.5% in low- and middle-income countries [29]. Interestingly, although the use of T&CM among cancer patients has increased worldwide from 1970 to 2018 [24,28], the subgroup analysis by the survey year indicates that the prevalence of T&CM use in Asian countries remains consistently high, suggesting a historically deep-rooted preference for T&CM. However, the substantial heterogeneity between studies suggests that these estimates may pose interpretive challenges, as the observed variation is likely due to factors beyond random chance [24]. Heterogeneity may arise from variations in data or study design, including differences in study population characteristics, data collection methods (such as whether an interview or self-completed questionnaire was used, and whether a predefined T&CM list or a free recall method was employed), the range of T&CM modalities covered, and the time period relevant to the survey questions on T&CM use [24,28,86].

The available data in this study imply the complexity and diversity of T&CM utilization. Asia is geographically vast and densely populated, with significant social and cultural variation. Each country’s healthcare system, including traditional medicine, is influenced by its unique historical and cultural context, and this accounts for the observed differences in T&CM utilization rates. The country-level data and subgroup analysis indicate that East Asia has a higher prevalence of T&CM use compared to other regions, consistent with previous studies reporting frequent T&CM use in East Asia [87]. However, given the substantial variations in economic development, cultural practices, and healthcare systems across Asia, our prevalence estimates may not fully capture the range of conditions present in different countries. This concern is highlighted by the fact that 29.3% of the included studies (12 studies) were conducted in Turkey, leading to uneven representation across the region [46,64,68,84]. Despite the substantial variation across studies, the need for rapid, evidence-based decisions in public health, combined with the surge in the scientific literature, necessitates research synthesis to address urgent issues [88]. Previous research has shown that patients using complementary therapies are more likely to refuse additional cancer treatment and have a higher risk of mortality [20], underscoring the need to identify factors associated with their use.

Health insurance coverage of T&CM is an emerging topic that reflects trends in T&CM utilization and serves as a significant factor influencing the high prevalence estimates of these practices among cancer patients, while also allowing for a national-level comparison of information [8]. T&CM utilization is higher in countries where health insurance covers T&CM, with the highest prevalence observed among cancer patients in countries with both national and private insurance coverage, such as China, India, Korea, and Taiwan. This finding supports previous research indicating that approximately one in two cancer patients with pain would be willing to use acupuncture for pain management if the treatment were covered by insurance [89]. Increasing the accessibility of evidence-based non-pharmaceutical treatments may be considered a national policy priority in response to public health issues, such as the opioid crises for pain or over-reliance on antibiotics [90]. Additionally, healthcare systems should offer a wide range of tools and necessary support, including T&CM, to enable doctors and patients to manage medical conditions effectively, with the goal of achieving universal health coverage (UHC) [91].

Furthermore, our study provides quantitative evidence that women with cancer are more likely to use T&CM, which is consistent with previous findings [28,29,56,63,78]. Gender is one of the most common demographic predictors of T&CM use among cancer patients [28]. Significant gender-based differences were also observed in subgroup analyses for East Asia, high-income countries, and countries with both state and private T&CM insurance coverage. This finding supports the results of a general population study indicating that women are three times more likely than men to utilize these approaches [92]. Explanations for the higher use of T&CM in female patients were that they generally utilize more health services than males [28,93] or seek health information more frequently [94]. However, healthcare access patterns across Asia can vary significantly from one region to another. A study investigating gender disparities in healthcare utilization in India revealed that men travel longer distances to reach a hospital than women, suggesting that women may encounter constraints in accessing healthcare services [95]. This gender disparity in healthcare utilization might drive women to seek T&CM outside conventional healthcare systems in Asia. Other frequently reported sociodemographic predictors of T&CM use included in this review were high household income [48,78,81] and the experience of symptoms [53,60,68].

Variability in survey conditions, such as differences in how T&CM modalities are defined and classified, contributes to the unexplained heterogeneity in studies of T&CM use [96,97,98]. In Asia, where T&CM has a long-standing history, it is often reported ambiguously whether participants used the modalities for medicinal purposes or general use. Some studies categorized herbs by specific names, such as stinging nettle and nigella, while others used broader terms like “herb”, “herbal tea”, or “herbal remedy”. Additionally, some studies did not provide usage percentages, merely listing T&CM modalities without further details. As a result, it was challenging to ascertain which types of T&CM were most commonly used. Addressing these issues requires ongoing global cooperation to standardize questionnaires on T&CM use. It is crucial to develop and implement questionnaires that account for the unique characteristics of each region, target population, and disease [99,100,101].

Consistent with previous research, the method of data collection—whether self-completed questionnaires or face-to-face interviews—affects the reported prevalence [24,28]. Although not statistically significant, the self-completed questionnaire method in this review yielded higher prevalence estimates compared to face-to-face interviews. Surveys conducted by highly trained interviewers might report a higher prevalence [24], but issues related to inaccurate reporting on sensitive topics also need to be addressed [102]. Additionally, studies conducted in conventional oncology settings might show reluctance among patients to disclose their T&CM use due to negative reactions from physicians [46,51,70]. Disclosure rates of T&CM use are notably lower among Asians [103] and Asian Americans [104] than in other regions.

The disclosure of T&CM use by cancer patients to their physicians remained stagnant, with rates continuing to be lower than the 50–60% reported in the prior review [23,105,106]. Despite the potential risks of interactions between T&CMs and conventional cancer therapies, this topic has only recently gained attention in Asia. About 70% of the studies addressing disclosure issues included in this review were published after 2011 [46,48,50,51,52,53,54,58,59,60,62,63,65,66,67,68,69,70]. However, there is a lack of research investigating predictors of non-disclosure, and only a few have reported physicians’ responses to disclosure [50,52,54,62,66,70,73,81]. Discussing T&CM with healthcare providers has been associated with improved patient care quality [107]. As physicians are typically the primary experts whom patients consult regarding cancer treatment, it is fundamental for them to be aware of their patients’ entire health histories to provide optimal care [106]. Nevertheless, discussions about T&CM during the cancer treatments are relatively infrequent, and when they do occur, they are most often initiated by patients. In line with previous studies [23,108], the lack of inquiry from doctors was identified as the primary reason for non-disclosure [46,48,52,59,63,68,69,70,73]. Our findings identified another significant reason that patients often feel no need to consult their doctors about T&CM use [46,50,62,63,68,70,79]. This may be due to the perception that T&CM is an inherent and widespread cultural value in Asia [107] and that family-oriented values exert a strong influence in Asia [109,110,111]. Most families are actively involved in the patient’s treatment process, regardless of the patient’s preferences, and frequently recommend traditional methods with which they are familiar or have heard about [111]. Moreover, the hierarchical nature of relationships, which is more pronounced in Asian societies, can influence doctor–patient communication dynamics [112]. Asian patients often experience doctor-centered communication styles, where critical attitudes toward T&CM use may discourage patients from discussing their use of these modalities with their physicians [113]. However, as levels of education and patient autonomy increase, Asian patients are increasingly seeking to participate in medical decision-making and engage in more interactive communication with their healthcare providers [112]. Consequently, there has been a proactive shift towards integrative cancer care in Asian countries such as Korea [114], China [115], Israel [116,117], and India [118], where active efforts are being made to meet the growing demand for T&CM among cancer patients.

Integrative oncology (IO) represents a paradigm shift in cancer care, emphasizing a holistic approach that considers individual patient preferences, needs, and values, while also evaluating the safety and efficacy of T&CM in conjunction with conventional cancer treatments [119,120]. However, for many Asian countries, integrative oncology remains an idealistic goal due to significant disparities in economic status and cultural backgrounds across the region. The credibility of T&CM, particularly in resource-limited areas, significantly impacts vulnerable patients, who may resort to unproven T&CM therapies without consulting a physician [121]. Additionally, a major challenge lies in the profound divergence between T&CM and biomedical healthcare systems [121]. Therefore, to effectively implement integrative cancer care in these Asian contexts, establishing multidisciplinary collaborations between biomedical and T&CM practitioners is essential. Open and meaningful discussions about T&CM between patients and physicians are also vital for fostering patient-centered care [122]. Although the studies reviewed did not examine specific factors influencing the disclosure of T&CM use, the findings suggest that patients’ relationships with healthcare providers likely affect their willingness to disclose such information [52]. Addressing these challenges may help in achieving the potential of integrative oncology to enhance cancer care outcomes in Asia.

This review has several limitations that need to be considered. First, meta-analyses on prevalence estimates often exhibit high I^2^ values and are susceptible to biases inherent to observational study designs [88,123]. Efforts were made to enhance the robustness of our estimation of T&CM use by identifying potential covariates that may contribute to the observed heterogeneity and stratifying the studies into more homogeneous subgroups. However, the substantial heterogeneity observed among the studies highlights the need for the cautious interpretation of the pooled prevalence of T&CM use and disclosure rate [24,46]. Second, the scarcity of data on the prevalence of T&CM use limits the generalizability of our findings. Data were lacking in many countries, particularly in two-thirds of the forty-seven countries classified within Asia. This may be attributed to the fact that English is not the native language of the Asian population. The studies adopted for this review were limited to articles published in English. This highlights the need for further research to fill this knowledge gap and provide a more comprehensive understanding of T&CM utilization among cancer patients in Asia. Third, the lack of specific data on the disclosure of T&CM use is a notable limitation. The reviewed literature reveals that the methods for defining and assessing T&CM disclosure varied significantly. While some studies examined the impact of healthcare providers’ communication styles on reasons for non-disclosure, this was addressed in only a few cases. Identifying determinants influencing disclosure, such as sociodemographic factors, remains underexplored. To address these gaps, more empirical research employing well-designed tools is necessary to better understand the reasons and factors associated with the non-disclosure of T&CM use.

Nonetheless, this review provides valuable insights. Although our review employed strategies similar to previous studies on T&CM use by cancer patients [23,24,28], its specific focus on the Asian region yields particularly promising findings. It is crucial to acknowledge that T&CM is often underestimated as a health resource despite its widespread use across Asia [8]. Our research has the potential to bridge existing evidence gaps and guide the integration of T&CM into cancer care in the region, ultimately enhancing patient outcomes and healthcare practices.

## 5. Conclusions

T&CM is widely used among cancer patients in Asia, with usage associated with demographic factors, geographical regions, and health insurance coverage. However, many patients do not disclosure their T&CM use to healthcare providers, often due to in-sufficient physician inquiry and a perceived lack of necessity. This underscores the critical need to improve patient–physician communication to ensure effective cancer treatment and patient well-being. Enhanced communication can improve medication adherence and pain management, thereby promoting patient-centered care. Future research should identify barriers to T&CM disclosure and their impact on treatment adherence, supporting a more integrative and patient-centered cancer care approach.

## Figures and Tables

**Figure 1 cancers-16-03130-f001:**
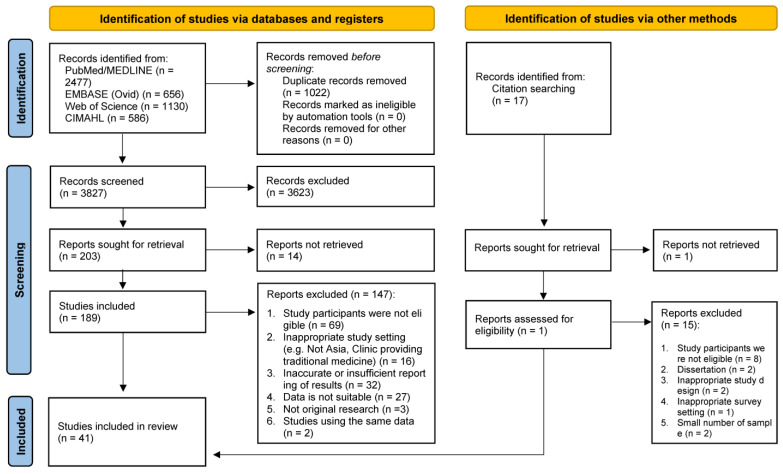
PRISMA flow diagram of study selection.

**Figure 2 cancers-16-03130-f002:**
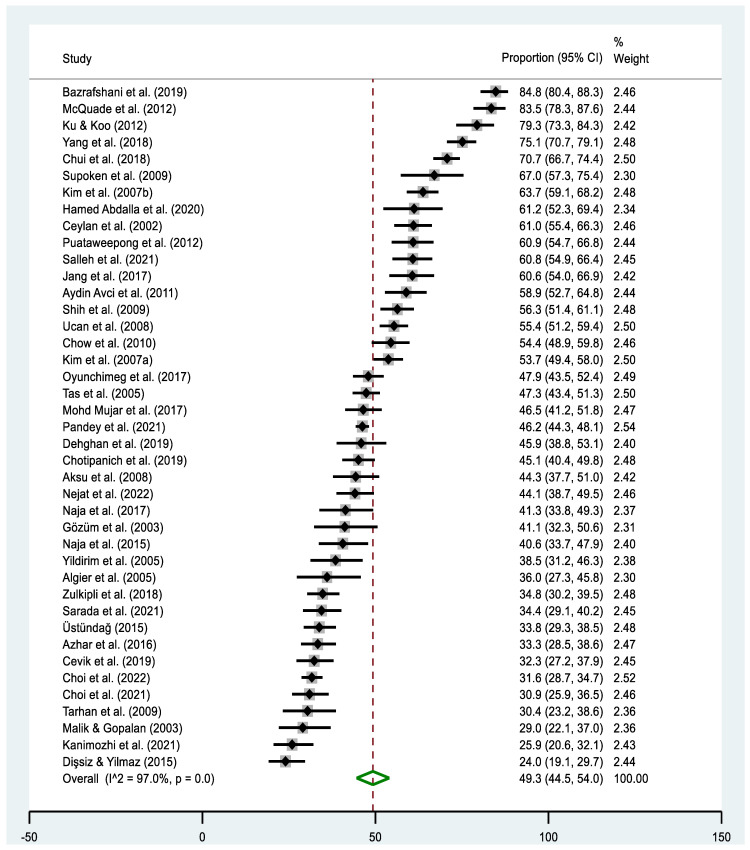
Forest plot of the pooled prevalence estimates of T&CM use among cancer patients in Asia. The black diamond dots represent the individual prevalence from each study, with the horizontal lines representing the 95% CI. The green rhombus and the red dashed line represent the pooled prevalence estimate and its 95% CI.

**Figure 3 cancers-16-03130-f003:**
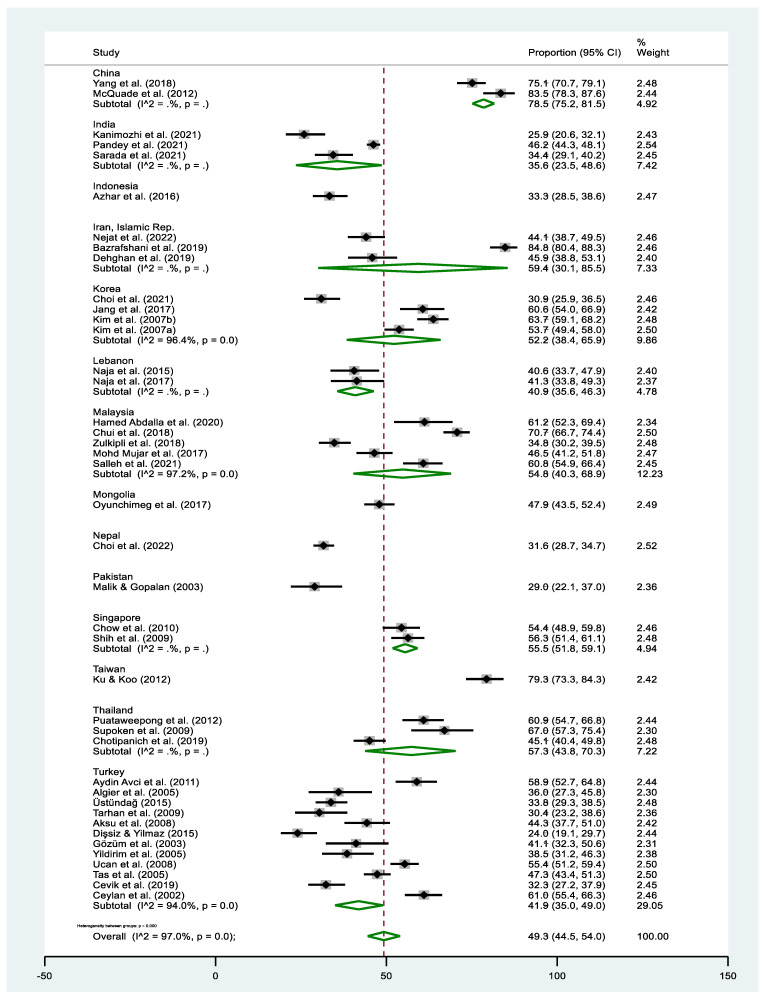
Forest plot of the pooled prevalence estimates of T&CM use among cancer patients in individual Asian countries. The black diamond dots represent the individual prevalence from each study, with the horizontal lines representing the 95% CI. The green rhombus and the red dashed line denote the pooled prevalence estimate and its 95% CI. Sub-groups with (I^2^ = . %, *p* = .) indicate that the number of studies were too few for the estimates to be calculated.

**Figure 4 cancers-16-03130-f004:**
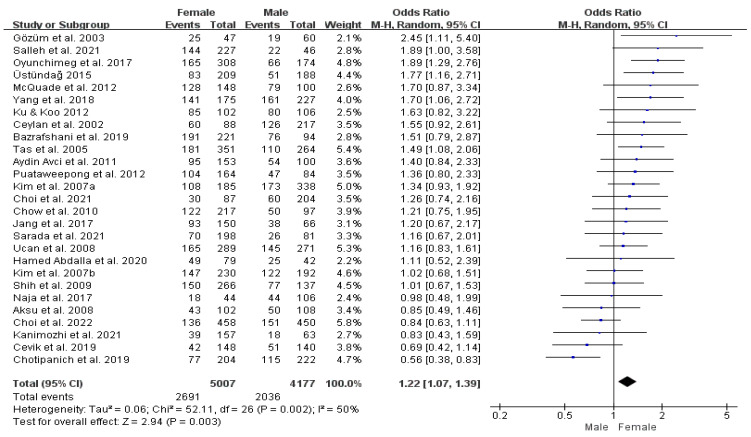
Forrest plot comparing female versus male prevalence of T&CM use. df = degrees of freedom; M-H = Mantel–Haenszel.

**Figure 5 cancers-16-03130-f005:**
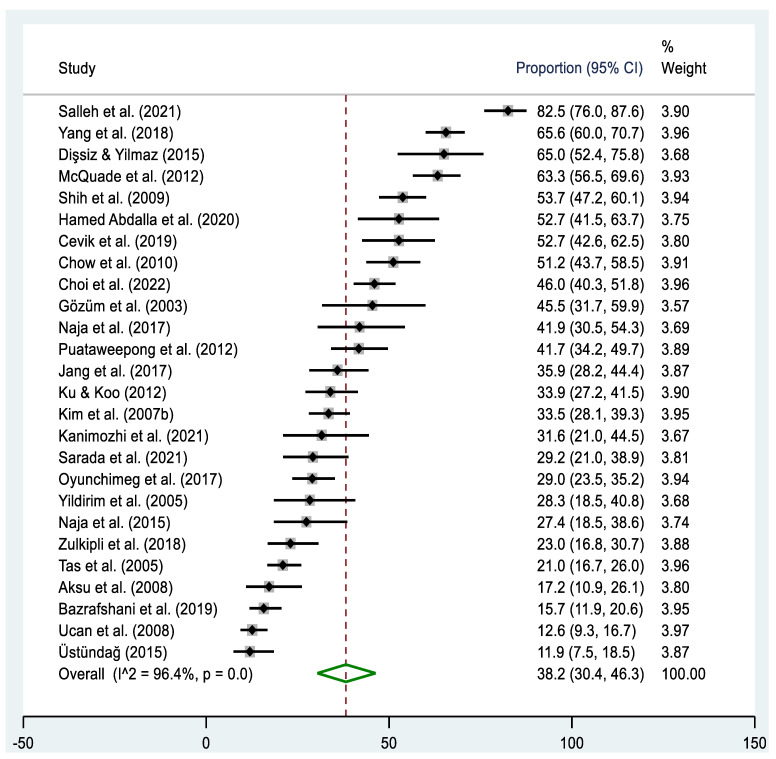
Forest plot of the pooled prevalence estimates of T&CM disclosure among cancer patients in Asia. The black diamond dots represent the individual prevalence from each study, with the horizontal lines representing the 95% CI. The green rhombus and the red dashed line represent the pooled prevalence estimate and its 95% CI.

**Table 1 cancers-16-03130-t001:** Summary and characteristics of each study investigating T&CM use among cancer patients (n = 41 studies).

First Author (Year)	Country ^†^	Study Design Setting Date of Study	Sample Size (Response Rate %)	Age Mean ± SD (Range)	Gender N (%)	Cancer Types N (%)	Prevalence of T&CM Use N (%)	Disclosure of T&CM Use N (%) ^‡^	Types of T&CM Used N (%)
Nejat et al. (2023) [45]	Iran	CSS, face-to-face interviews Ayatollah Khansari Center NR	320 (NR)	55.11 ± 15.6	M: 161 (50.3) F: 159 (49.7)	HM: 83 (25.9) Br: 43 (13.4) Col: 37 (11.5) Lu: 31 (9.7) GI: 22 (6.9) Bone: 22 (6.9) HB:19 (5.9) GU: 17 (5.3) O: 47 (14.6)	141 (44.1)	NR	Visiting holy places Yoga Prayer therapy Medicinal plants Special diets
Choi et al. (2022) [46]	Nepal	CSS using face-to-face interviews Two tertiary hospitals Between December 2018 and August 2019	908 (97.6)	53.7 ± 15.6 (18–92)	M: 450 (49.6) F: 458 (50.4)	GI: 207 (26.4) HN: 184 (23.5) HM: 166 (21.2) Br + Gy: 102 (13.0) GU: 72 (9.2) Lu: 53 (6.8)	287 (31.6)	132 (46.0)	Ayurveda: 119 (46.5) Yoga: 83 (32.4) Herbal products: 79 (30.9) Meditation: 74 (28.9) Honey: 60 (23.4) Ginger: 58 (22.7)
Choi et al. (2021) [47]	Korea	A prospective survey-based study using medical charts and self-completed questionnaires Seven medical centers Between August and October 2019	291 (NR)	NR	M: 204 (70.1) F: 87 (29.9)	Lu: 291 (100.0)	90 (30.9)	NR	Mushrooms Oncothermia Ginseng High-dose vitamin C
Kanimozhi et al. (2021) [48]	India	CSS with self-completed questionnaires A tertiary care teaching hospital June 2016–February 2017	220 (NR)	51.8 ± 12.3 (18–80)	M: 63 (28.6) F: 157 (71.4)	Br: 69 (31.4) GI: 59 (26.9) Gy: 31 (14.1) HN: 19 (9.7) Lu: 12 (5.5) O: 30 (13.6)	57 (25.9)	18 (31.6)	Self-prepared folklore medicines: 39 (66.7) Herbal preparations prescribed by traditional medicine practitioners: 16 (28.1) Herbal products from the market: 3 (5.3)
Pandey et al. (2021) [49]	India	A questionnaire-based study, interviews From January to December 2018	2614 (NR)	52.3 ± 10.4	NR	NR	1190 (46.2)	NR	Ayurveda: 428 (35.9) Yoga/Naturopathy: 381 (32.0) Homeopathy: 143 (12.0) Unani: 71 (5.9) Others: 167 (14.0)
Salleh et al. (2021) [50]	Malaysia	CSS using interviews Three departments (surgery, medicine, and gynecology) at a local hospital Between September and November 2016	273 (NR)	48 ± 12.34	M: 46 (16.8) F: 227 (93.2)	Gy: 128 (46.9) Br: 78 (28.5) Col: 54 (19.8) O: 13 (4.8)	166 (60.8)	137 (82.5)	Dietary supplements (100%) Herbal products (92.8%) Traditional Malay therapy
Sarada et al. (2021) [51]	India	CSS; face-to-face interviews From January 2019 to December 2019	279 (87.0)	55.99 ± 11.53 (users) 55.53 ± 11.88 (non-users)	M: 81 (29.0) F: 198 (71.0)	Br: 122 Gy: 36 HM: 27 Col: 27 Lu: 12 GI: 10 GU: 7 O: 38	96 (34.4)	28 (29.2)	Home remedy: 35 Folk medicine: 34 Nutraceuticals: 22 Ayurveda: 20 Homeopathy: 13 Yoga: 5
Hamed Abdalla et al. (2020) [52]	Malaysia	The observational CSS using a self-administered questionnaire The outpatient clinic at the university Hospital From July to October 2017	121 (40.6)	55.2 ± 14.3	M: 42 (34.7) F: 79 (65.3)	Br: 46 (38.0) HM: 23 (19.0) Col: 16 (13.2) Lu: 8 (6.6) HN: 7 (5.8) GI: 7 (5.8) O: 14 (11.6)	74 (61.2)	39 (52.7)	Vitamins: 45 (25.6) Islamic Medical Practice: 25 (14.2) Nutritional Therapy: 20 (11.4) Malay Herbs: 15 (8.5)
Bazrafshani et al. (2019) [53]	Iran	CSS using medical charts and face-to-face interviews Two public health sectors and one private health sector From February to August 2017	315 (NR)	51.2 ± 14.0 (18–92)	M: 94 (29.8) F: 221 (70.2)	Br: 119 (37.8) GI: 60 (19.0) HM: 56 (17.8) Gy: 32 (10.2) Lu: 17 (5.4) O: 31 (9.9)	267 (84.8) *	42 (15.7) *	Herbal medicine: 267 (84.1)
Cevik et al. (2019) [54]	Turkey	A descriptive, relational CSS using a researcher-administered questionnaire Two outpatient clinics in chemotherapy units From November 2014 to May 2015	288 (80.0)	55.9 ± 14.5 (18–90)	M: 140 (48.6) F: 148 (51.4)	GI: 86 (29.8) Br: 69 (24.0) Lu: 66 (22.9) GU: 30 (10.5) HN: 20 (6.9) HM: 17 (5.9)	93 (32.3)	49 (52.7)	Herbal: 45 (48.4) Honey: 28 (30.1) Religious practices: 27 (29.0)
Chotipanich et al. (2019) [55]	Thailand	CSS using face-to-face interviews and medical record reviews A referral cancer hospital From May to December 2018	426 (NR)	NR	M: 222 (52.1) F: 204 (47.9)	HN: 167 (39.2) Gy: 71 (16.7) Col: 67 (15.7) Br: 55 (12.9) O: 66 (15.5)	192 (45.1)	NR	Unlabeled fresh and processed herbal products: 74 (34.3) Ya Mor Sang: 41 (20.0) Lingzhi mushroom: 35 (16.2)
Dehghan et al. (2019) [56]	Iran	CSS using a self-reported questionnaire Cancer clinic and Yas Association of Kerman From April 2016 to January 2017	181 (70.0)	49.6 ± 16.9 (18–87)	M: 88 (48.6) F: 93 (51.4)	HM: 69 (38.1) GI: 27 (14.9) Genital cancers: 25 (13.8) O: 60 (33.2)	83 (45.9)	31 (37.3)	Prayer: 167 (92.3) Vow (nazr): 154 (85.1) Herbal medicine: 80 (44.2) Wet cupping: (Hijama) 6 (3.3) Dry cupping: 5 (2.8) Massage: 4 (2.2) Leech therapy: 4 (2.2) Acupuncture: 1 (0.6) Acupressure: 1 (0.6)
Chui et al. (2018) [57]	Malaysia	CSS using a researcher-administered questionnaire Two outpatient chemotherapy referral centers From March 2012 to August 2013	546 (78.1)	NR	F: 546 (100.0)	Br: 546 (100.0)	386 (70.7)	NR	Prayer: 43 (86.0) Traditional healer: 12 (92.3) Massage: 10 (20.0) Vitamin and mineral supplement: 8 (26.9)
Yang et al. (2018) [58]	China	Two parallel CSS using a self-administered questionnaire Military Medical University affiliated hospital From July to December 2015	402 (39.9)	56.1 ± 10.8	M: 227 (56.5) F: 175 (43.5)	Col: 110 (27.4) GI: 85 (21.1) Br: 52 (12.9) Es: 43 (10.7) Lu: 32 (8.0) HB: 22 (5.5) Gy: 13 (3.2) HM: 4 (1.0) O: 41 (10.2)	302 (75.1)	198 (65.6)	Proprietary Chinese medicine: 224 (74.2) Chinese herbal medicine: 132 (43.7) Dietary therapy: 51 (16.9) Acupuncture: 22 (7.3) Tai chi: 12 (4.0) Chi gong: 6 (2.0) Massage therapy: 5 (1.7) Other: 6 (2.0)
Zulkipli et al. (2018) [59]	Malaysia	CSS using individually interviews A tertiary teaching hospital having a breast cancer registry system From February 2012 to December 2014	400 (69.7)	57.0 ± 16.0	F: 400 (100.0)	Br: 400 (100)	139 (34.8)	32 (23.0) *	Dietary supplements: 107 (77.0) Spiritual/Prayer: 40 (28.8) Traditional Chinese medicine: 32 (23.0)
Jang et al. (2017) [60]	Korea	CSS using face-to-face interviews A cancer center in Korea NR	216 (NR)	59.0 ± 11.6	M: 66 (30.6) F: 150 (69.4)	Gy: 67 (31.0) GI: 52 (24.1) Br: 36 (16.7) HN: 26 (12.0) Lu: 20 (9.3) GU: 12 (5.6) O: 3 (1.3)	131 (60.6)	36 (27.5)	Herbal medicine: 89 (67.9) Vitamins: 71 (54.2) Acupuncture: 49 (37.4) Fatty acids: 26 (19.8) Minerals: 24 (18.3)
Mohd Mujar et al. (2017) [61]	Malaysia	Multi-center cross-sectional study using medical files and face-to-face interviews Six public hospitals From January to December 2012	340 (39.1)	53 (23–74)	F: 340 (100.0)	Br: 340 (100.0)	158 (46.5)	NR	Multivitamins: 108 Prayers: 53 Traditional Chinese medicine: 38 Special diet (herbs, juices): 12 Cupping: 12
Naja et al. (2017) [62]	Lebanon	CSS using face-to-face interviews Outpatient clinics at a teaching hospital From September 2015 to August 2016	150 (96.2)	NR	M: 106 (71.0) F: 44 (29.0)	Lu: 150 (100.0)	62 (41.3)	26 (41.9)	Dietary supplements (79.0) Herbal remedies (27.0) Vitamin/minerals (15.0)
Oyunchimeg et al. (2017) [63]	Mongolia	CSS using face-to-face interviews National cancer center From September 2015 to February 2016	482 (95.6)	58 (21–89)	M:174 (36.1) F: 308 (63.9)	GI: 174 (36.1) GU: 140 (29.0) Br: 75 (15.6) O: 93 (19.3)	231 (47.9)	67 (29.0)	Tripe (goat, sheep): 86 (37.2) Mantra: 46 (19.9) Rhubarb: 42 (18.2) Wild animal products: 42 (18.2)
Azhar et al. (2016) [64]	Indonesia	A questionnaire survey with interviews West Java Cancer Center in Bandung From July 2014 to July 2015	330 (NR)	45	F: 330 (100.0)	Br: 330 (100.0)	110 (33.3)	NR	NR
Dişsiz and Yilmaz (2016) [65]	Turkey	CSS using face-to-face interviews Outpatient chemotherapy unit 2014	250 (78.1)	55.0 ± 11.2	M: 88 (35.2) F: 162 (64.8)	Br: 95 (38.0) Col: 61 (24.4) Gy: 29 (11.6) GI: 17 (6.8) Lu: 11 (4.4) HB: 10 (4.0) GU: 8 (3.2) O: 19 (7.6)	60 (24.0)	39 (65.0)	Herbs and folk remedies: 23 (32.6) Honey/royal jelly: 20 (28.2) Kefir: 7 (9.8) Vitamins: 2 (3.0) Shark cartilage: 2 (3.0)
Naja et al. (2015) [66]	Lebanon	CSS using face-to-face interviews (1st) and telephone surveys (2nd) Two major health care facilities From October 2013 to August 2014	180 (94.7)	53.8 ± 9.9	M: 88 (35.2) F: 162 (64.8)	Br: 180 (100.0)	73 (40.6)	20 (27.4)	Special foods (honey, black seed, camel milk, soy, pomegranate, and ginger) Herbal teas Diet supplements (prebiotic and graviola pills)
Üstündağ et al. (2015) [67]	Turkey	CSS; medical files and face-to-face interviews Daytime chemotherapy unit From January to June 2013	397 (75.3)	53.7 ± 12.8 (20–92)	M: 188 (47.4) F: 209 (52.6)	GI: 90 (22.6) Br: 88 (22.1) Col: 79 (19.9) HM: 50 (12.6) Lu: 30 (7.6) GU: 30 (7.6) HN: 11 (2.8) O: 14 (3.6)	134 (33.8)	16 (11.9)	Religious and cultural rituals: 361 (92.2) Nutritional and herbal products: - Honey: 43 (32.0) - Stinging nettle: 30 (22.3) - Nigella: 27 (20.1) - Grape seed: 19 (14.1)
Ku and Koo (2012) [68]	Taiwan	CSS using face-to-face interviews Outpatient clinics at the medical center From April to July 2007	208 (NR)	55.2 ± 13.0 (18–88)	M: 106 (51.0) F: 102 (49.0)	Br: 52 (25.0) Lu: 28 (13.5) GI: 28 (13.5) HN: 27 (13.0) Col: 25 (12.0) GU: 24 (11.5) O: 24 (11.5)	165 (79.3)	109 (66.1)	Prescribed Chinese medicine: 70 (33.7) Vitamins: 59 (28.4) Chanting: 50 (24.0) Enzyme therapy: 38 (18.3) Agaricus subrufescens: 32 (15.4) Herbal remedy: 30 (14.4)
McQuade et al. (2012) [69]	China	CSS using a self-administered questionnaire Outpatient clinics at the university cancer center June 2005	248 (50.9)	NR	M: 100 (40.3) F: 148 (59.7)	Br: 78 (31.4) GI: 57 (22.9) HM: 29 (11.7) HN: 27 (10.8) Lu: 25 (9.9) O: 33 (13.5)	207 (83.5)	NR (63.5)	Tonics/supplements (50.4) Food therapy (41.5) Herbal decoctions (47.6) Patent medicines (31.8) Tai Qi (4.0) Acupuncture (1.3)
Puataweepong et al. (2012) [70]	Thailand	CSS using face-to-face interviews Radiotherapy outpatient clinic at the Ramathibodi hospital From June to July 2011	248 (NR)	53.7 (user) 54.3 (non–user)	M: 84 (33.9) F: 164 (66.1)	Br: 62 (25.0) GU: 53 (21.4) HN: 51 (20.6) GI: 23 (9.3) Brain: 15 (6.0) Lu: 14 (5.6) O: 30 (12.1)	151 (60.9)	63 (41.7)	Food/vitamin supplement: 86 (56.9) Dietary adjustment: 75 (49.7) Meditation: 64 (42.4) Herbal medicine: 47 (31.1)
Aydin Avci et al. (2011) [71]	Turkey	A descriptive study using a self-administered descriptive questionnaire University hospital From March to June 2008	253 (90.0)	53.8 ± 13.6 (≥18)	M:100 (39.5) F: 153 (60.5)	GI: 107 (42.3) Br: 63 (24.9) Lu: 33 (13.0) HM: 27 (11.0) GU: 16 (6.3) HN: 7 (2.8)	149 (58.9)	NR	Herbal treatment: 70 (27.7) Self-prayer: 64 (25.3) Fast walking: 62 (24.5) Music therapy: 38 (15.0) Imagery: 37 (14.6) Diets: 34 (13.4) Psychotherapy: 31 (12.3) Multivitamins: 28 (11.1)
Chow et al. (2010) [72]	Singapore	CSS using a standardized, interviewer-administered questionnaire Dept. of Radiation Oncology of the National University Cancer Institute January 2007	316 (77.1)	55 (18–99)	M: 97 (30.7) F: 217 (68.7) Missing: 2	Br: 100 (31.6) HM: 48 (15.2) Col: 43 (13.6) Lu: 25 (7.9) O: 100 (31.6)	173 (54.7)	88 (50.9)	Traditional Chinese medicine: (68.8) Health supplements: (52.6) Taiji/Qi Gong: (12.1) Reflexology: (6.4) Other oral CAMs: (5.8) Acupuncture/Moxibustion: (3.5) Other non-oral CAMs: (2.9)
Shih et al. (2009) [73]	Singapore	CSS; an interviewer-administered questionnaire The ambulatory treatment unit of national cancer center From October 2007 to March 2008	403 (NR)	Median 56 (22–84)	M: 137 (34.0) F: 266 (66.0)	Br: 141 (35.0) Lu: 64 (15.9) HN: 56 (13.9) Col: 47 (11.7) GU: 47 (11.7) HM: 10 (2.5) O: 38 (9.4)	227 (56.3)	122 (53.7)	Food supplements (bird’s nest, essence of chicken): (59.0) TCMs: (48.5) Special diet (organic vegetables and fruit): (40.5) Vitamins: (39.6) Minerals: (20.7)
Supoken et al. (2009) [74]	Thailand	CSS using face-to-face interviews NR From October to December 2008	100 (NR)	50.1 (21–69)	F:100 (67.0)	Gy: 100 (100.0)	67 (67.0)	NR	Buddhist praying: 62 (92.5) Herbal medicines: 27 (40.3) Exercises: 25 (37.3) Diet modifications: 16 (23.9) Thai massage: 12 (17.9) Dietary supplements: 5 (7.5)
Tarhan et al. (2009) [75]	Turkey	CSS using medical files and face-to-face interviews Medical oncology outpatient clinic of the training and research hospital From April to June 2008	135 (NR)	(≥30)	F: 135 (100)	Br: 135 (100.0)	41 (30.4)	NR	Herbal remedies: 40 (97.6) Mind–body interventions: 1 (2.4)
Aksu et al. (2008) [76]	Turkey	CSS a using self-administered questionnaires Radiation oncology department From November 2005 to June 2006	210 (NR)	Mean: 52.6 Median: 54	M: 108 (51.4) F: 102 (48.6)	Br: 59 (28.1) Lu: 47 (22.4) GI: 21 (10.0) GU: 18 (8.6) Gy: 17 (8.1) HN: 15 (7.1) HM: 10 (4.8) O: 23 (11.0)	93 (44.3)	16 (17.2) *	Stinging nettle (63.4) Herbal teas (23.7) Vitamin combinations (20.4) Green tea (15.1) Flax seed (12.9)
Ucan et al. (2008) [77]	Turkey	A descriptive survey using questionnaire/interview Outpatient clinics of medical Oncology, University From April to October 2006	560 (NR)	49.5 ± 15.8 (18–76)	M: 271 (48.4) F: 289 (51.6)	HM: 173 (30.9) GI: 124 (22.1) Lu: 102 (18.2) GU: 75 (13.4) Br: 67 (12.0) O: 19 (3.4)	310 (55.4)	39 (12.6) *	Herbal therapy or herbal essences (82.9) - Nettle (55.4) - Pomegranate juice (14.8) - Raisins (13.5)
Kim et al. (2007a) [78]	Korea	A longitudinal study using face-to-face interviews The National Cancer Center 2003–2005	523 (96.7)	NR	M: 338 (64.6) F: 185 (35.4)	Col: 219 (41.9) GI: 187 (35.7) HB: 117 (22.4)	281 (53.7)	NR	NR
Kim et al. (2007b) [79]	Korea	CSS using a researcher-administered questionnaire Inpatients and outpatients’ clinics of 9 hospitals From August to September 2002	422 (98.4)	NR	M: 192 (45.5) F: 230 (54.5)	GI: 168 (39.8) Br: 87 (20.6) Lu: 63 (14.9) HB: 53 (12.6) Gy: 51 (12.1)	269 (63.7)	90 (33.5)	Mushroom: 137 (50.9) Vegetable extract: 44 (16.4) Traditional herbal medicine: 42 (15.6) Ginseng: 41 (15.2) Meditation or prayer: 34 (12.6) Multi-vitamin: 21 (7.8)
Algier et al. (2005) [80]	Turkey	A descriptive cross-sectional survey using face-to-face interviews Outpatient clinics of a state oncology hospital and in the oncology inpatient and outpatient clinics of a university hospital From June to August 2003	100 (NR)	NR	M: 57 (57.0) F: 43 (43.0)	HN: 23 (23.0) Lu: 18 (18.0) Col: 17 (17.0) Br: 12 (12.0) GI: 9 (9.0) HM: 4 (4.0) Gy: 3 (3.0) GU: 2 (2.0) O: 12 (12.0)	36 (36.0)	NR	Herbal therapy or herbal essences: 32 (88.9) - Nettle or seed: 18 (57.6) - Thyme 4 (10.2) - Chamomile: 3 (8.5)
Tas et al. (2005) [81]	Turkey	CSS using self-administered questionnaires, occasionally face-to-face interview Department of Medical Oncology at the Institute of Oncology, University of Istanbul. From July to October 2001	615 (97.6)	Median: 53 (18–82)	M: 264 (42.9) F: 351 (57.1)	Br: 157 (25.5) GI: 138 (22.4) HM: 82 (13.3) Lu: 67 (10.9) Gy: 59 (9.6) GU: 37 (6.0) O: 75 (12.2)	291 (47.3)	61 (21.0) *	Herbal agents: 276 (94.9) Nettle +/− honey: 162 (55.7) Nettle + other herbal agents: 94 (32.3) Other herbal agents: 20 (6.9) Non-herbal agents: 6 (2.1) Combined: 9 (3.1)
Yildirim et al. (2005) [82]	Turkey	CSS; face-to-face interviews Ministry of Health Aegean Obstetrics and Gynecology Teaching Hospital Department of Gynecologic Oncology Between December 2002 and March 2005	156 (51.8)	49.4 ± 3.9 (34–63)	F: 156 (100)	GI: 156 (100)	60 (38.5)	17 (28.3)	Herbal medicine Supplement Exercise Mind–body techniques Massage Others
Gözüm et al. (2003) [83]	Turkey	CSS using medical charts and face-to-face interviews Radiation oncology department of the hospital From January to October 2001	107 (NR)	55.6 ± 12.4	M: 60 (56.1) F: 47 (43.9)	GI: 30 (28.0) Br: 25 (23.4) HN: 25 (23.4) Lu: 21 (19.6) GU: 6 (5.6)	44 (41.1)	20 (45.5)	Herb (oral): 42 (95.5) Herb (ointment): 2 (4.5)
Malik and Gopalan (2003) [84]	Pakistan	NR, using interviews Oncology service at the National Cancer Institute From September 1997 to February 1999	138 (NR)	46.1	F: 138 (100.0)	Br: 138 (100.0)	40 (29.0)	NR	Homeopathy (70.0) Spiritual therapy, including visits to spiritual healers and tombs (15.0) Ayurvedic medicine (13.0) Herbal medicines (2.0)
Ceylan et al. (2002) [85]	Turkey	CSS using face-to-face interviews Inpatients; the departments of Hematology and Oncology at the Military Medical Academy (GMMA) From December 1997 to December 1998	305 (93.6)	NR	M: 217 (71.1) F: 88 (28.9)	HM: 138 (45.3) GU: 26 (8.5) Lu: 24 (7.9) Br: 20 (6.6) O: 97 (31.8)	186 (61.0)	NR	Herbal preparations: 133 (71.5) Herbal preparations + religious practices: 21 (11.3) Religious practices: 15 (8.0) “Old woman medicine” practices 8 (4.3)

* The percentage values were recalculated due to a calculation error. Abbreviations: CSS: Cross-sectional study, Br: Breast cancer, Col: Colorectal cancer, GI: Gastrointestinal cancer, GU: Genitourinary cancer, Gy: Gynecological cancer, Lu: Lung cancer/Respiratory/Thorax cancer, HB: Hepatobiliary/Liver/Pancreas cancer, HM: Hematologic cancer (including leukemia, lymphoma, and multiple myeloma), HN: Head and neck cancer, ES: Esophagus, O: Others, NR: not reported; ^†^ The 14 Asian countries included in the study are categorized by geographical region and the number of studies included in each country as follows: West Asia (Iran (n = 3), Lebanon (n = 2), and Turkey (n = 12)), South Asia (India (n = 3), Nepal (n = 1), and Pakistan (n = 1)), Southeast Asia (Indonesia (n = 1), Malaysia (n = 5), Singapore (n = 2), and Thailand (n = 3)), and East Asia (Korea (n = 4), China (n = 2), Mongolia (n = 1), and Taiwan (n = 1)). ^‡^ Data on disclosure to doctors about T&CM use were extracted from the variable labeled “Disclosure to Doctors” (including discuss, report, inform, ask, physician consultation, communication with doctors, etc.).

**Table 2 cancers-16-03130-t002:** Subgroup analyses by characteristics of the difference in the pooled prevalence of T&CM use.

Categories		k **	Pooled Prevalence of T&CM Use (95% Cis ^†^)	Test of Heterogeneity	Random Test for Heterogeneity between Subgroups		k **	Pooled T&CM Disclosure Rate (95% Cis ^†^)	Test of Heterogeneity	Random Test for Heterogeneity between Subgroups	
				I^2^ (%)	Q	*p*-value			I^2^ (%)	Q	*p*-value
Overall pooled prevalence estimates		41	49.3 (44.5–54.0)	97.0	1326.5	<0.001	26	38.2 (30.4–46.3)	96.3	703.5	<0.001
Year of survey					4.200	0.040 *				1.310	0.252
	2014–2019	19	44.2 (37.4–51.2)	97.4			13	42.9 (30.7–55.6)	96.7		
	1998–2013	22	53.7 (47.9–59.4)	95.5			13	33.6 (24.1–43.8)	95.9		
Sample size calculation					0.073	0.788				0.099	0.753
	Yes	8	47.8 (36.4–59.4)	97.4			4	36.4 (26.7–46.7)	78.8		
	No	33	49.6 (44.3–54.9)	96.0			22	38.6 (29.6–48.0)	96.9		
Data collection					0.409	0.522				0.854	0.355
	Face-to-face interview	33	48.2 (43.4–53.0)	96.6			21	36.4 (28.4–44.8)	96.0		
	Self-completed questionnaire	8	53.6 (38.0–68.9)	97.9			5	45.9 (28.0–64.5)	95.7		
Geographical regions of Asia ^†^					17.714	0.001 *				6.201	0.102
	Western Asia	17	44.9 (37.4–52.4)	96.2			11	29.0 (20.1–38.7)	93.3		
	South Asia	5	33.5 (25.1–42.6)	96.0			3	36.2 (24.6–48.6)	–		
	Southeast Asia	11	53.6 (45.7–61.5)	95.5			6	51.1 (34.6–67.4)	96.2		
	East Asia	8	62.4 (50.5–73.7)	97.6			6	43.4 (29.8–57.6)	96.3		
Country classifications by income level ^‡^					3.085	0.214				2.694	0.260
	High-income economies	6	53.3 (44.5–62.0)	94.2			4	43.5 (33.0–54.3)	89.3		
	Upper-middle-income economies	25	51.0 (44.4–57.6)	96.8			17	39.5 (28.0–51.7)	97.2		
	Lower-middle-income economies	10	42.5 (33.4–51.8)	97.7			5	29.8 (18.4–42.6)	93.6		
Health insurance covers T&CM ^§^					46.242	<0.001					
	National health insurance	9	53.5 (42.8–64.0)	96.4							
	Private health insurance	19	46.7 (40.6–52.9)	95.4							
	Both	10	55.8 (44.7–66.6)	98.2							
	None	3	31.7 (29.3–34.2)	-							
Gender					0.248	0.618					
	Male	27	50.1 (44.3–56.0)	92.9	368.084	<0.001					
	Female	36	52.3 (46.3–58.2)	96.2	920.714	<0.001					
Types of cancer ^#^											
	Breast cancer	29	56.8 (48.8–64.6)	95.7	650.639	<0.001					
	Lung cancer	21	51.9 (44.4–59.4)	81.9	110.255	<0.001					
	Gastrointestinal cancer	18	55.7 (48.3–62.9)	87.6	136.659	<0.001					
	Genitourinary cancer	14	55.2 (44.3–65.9)	76.2	54.638	<0.001					
	Head and neck cancer	13	54.9 (43.8–65.9)	80.8	62.658	<0.001					
	Hematological cancer	12	58.5 (42.6–73.7)	92.9	154.822	<0.001					
	Gynecological cancer	12	55.8 (42.8–68.4)	90.4	114.745	<0.001					
	Colorectal cancer	9	60.6 (46.7–73.7)	91.4	92.399	<0.001					
	Hepatobiliary cancer	3	64.3 (57.2–71.0)	–							

* *p* < 0.1 (The null hypothesis is rejected; heterogeneity is present), I^2^: The variation in ES is attributable to heterogeneity; ** k: Number of prevalence estimates, CIs: Confidence intervals. ^†^ The 14 Asian countries included in the study were categorized by geographical region and the number of studies included in each country as follows: West Asia (Iran (n = 3), Lebanon (n = 2), and Turkey (n = 12)), South Asia (India (n = 3), Nepal (n = 1), and Pakistan (n = 1)), Southeast Asia (Indonesia (n = 1), Malaysia (n = 5), Singapore (n = 2), and Thailand (n = 3)), and East Asia (Korea (n = 4), China (n = 2), Mongolia (n = 1), and Taiwan (n = 1)). ^‡^ The income level classifications (2021–2022) used in this study were obtained from the World Bank: high-income economies (>USD 12,695), upper-middle-income economies (USD 4096–USD 12,695), and lower-middle income economies (USD 1046–USD 4095); ^§^ The countries where health insurance covers T&CM are categorized based on data from the WHO as follows (WHO global report on traditional and complementary medicine 2019: World Health Organization, 2019): (1) National health insurance (Iran, Lebanon, Thailand, Mongolia), (2) private health insurance (Turkey, Malaysia, Singapore), and (3) both (China, India, Korea, and Taiwan). T&CMs are not covered by health insurance in Indonesia, Nepal, and Pakistan. ^#^ Others are not included.

**Table 3 cancers-16-03130-t003:** Cancer-specific pooled prevalence estimates of T&CM by economic income level and region.

	The Pooled Prevalence Estimates of T&CM Use (95% CIs)								Random Test for Heterogeneity between Sub-Groups	
Types of Cancer	k **	Overall Pooled Estimates (95% CIs)	k	High-Income Economies ^†^ (n = 5)	k	Upper-Middle-Income Economies ^†^ (n = 22)	k	Lower-Middle-Income Economies ^†^ (n = 6)	Q	*p*-Value
Breast cancer	29	56.8 (48.8–64.6)	4	58.8 (53.7–63.9)	19	58.3 (48.3–68.0)	6	50.1 (28.7–71.4)	0.577	0.750
Lung cancer	21	51.9 (44.3–59.4)	5	46.2 (32.8–59.8)	13	54.9 (45.3–64.3)	5	48.9 (18.5–79.7)	1.074	0.585
Gastrointestinal cancer	18	55.7 (48.3–62.9)	3	63.2 (58.4–67.9)	12	51.7 (41.9–61.4)	3	62.6 (39.1–83.5)	4.584	0.101
Genitourinary cancer	14	55.2 (44.3–65.9)	1	66.7 (39.1–86.2)	11	51.8 (39.3–64.2)	2	58.5 (51.8–65.1)	1.491	0.475
Head and neck cancer	13	54.9 (43.8–65.9)	1	53.8 (35.5–71.2)	10	55.4 (39.7–70.5)	2	61.4 (54.4–68.1)	0.929	0.628
Hematological cancer	12	58.5 (42.6–73.7)	1	45.8 (32.6–59.7)	9	51.5 (36.3–66.5)	2	85.8 (80.8–90.1)	44.199	<0.001
Gynecological cancer	12	55.8 (42.8–68.4)	2	65.3 (56.3–73.7)	8	51.8 (38.7–64.8)	2	58.4 (45.8–70.5)	3.092	0.213
Colorectal cancer	9	60.6 (46.7–73.7)	3	50.4 (40.0–60.9)	6	65.2 (44.8– 83.1)	–	–	1.598	0.206
Hepatobiliary cancer	3	64.3 (57.2–71.0)	2	64.2 (56.8–71.3)	1	63.6 (43.0–80.3)	–	–	0.008	0.928
**Types of Cancer**	**k**	**East Asia (n = 8)**	**k**	**Southeast Asia (n = 11)**	**k**	**South Asia (n = 3)**	**k**	**Western Asia (n = 12)**	**Q**	** *p* ** **-Value**
Breast cancer	6	75.0 (60.3–87.3)	10	55.7 (42.3–68.6)	3	40.6 (10.8–74.9)	10	51.6 (39.7–63.5)	7.179	0.066 *
Lung cancer	5	60.7 (39.7–80.0)	4	57.7 (46.2–68.8)	2	35.1 (23.5–47.5)	9	47.4 (38.6–56.3)	8.291	0.040 *
Gastrointestinal cancer	7	61.2 (51.8–70.2)	2	46.8 (28.3–65.7)	1	61.8 (54.8–68.5)	8	49.7 (36.3–63.1)	4.540	0.209
Genitourinary cancer	4	62.9 (45.5–78.8)	2	75.5 (57.6–90.7)	1	77.8 (66.4–86.7)	7	41.8 (31.4–52.5)	23.768	<0.001
Head and neck cancer	3	80.2 (54.0–97.5)	3	54.6 (44.2–64.8)	2	61.4 (54.4–68.1)	5	32.8 (22.2–44.2)	21.236	<0.001
Hematological cancer	2	91.1 (76.3–99.7)	2	49.3 (37.5–61.1)	1	86.7 (80.6–91.5)	7	46.6 (30.0–63.6)	50.463	<0.001
Gynecological cancer	3	71.2 (57.3–83.4)	4	48.2 (26.6–70.1)	1	16.1 (5.5–33.7)	4	59.6 (33.4–83.3)	25.451	<0.001
Colorectal cancer	3	69.9 (39.0–93.4)	5	60.6 (45.2–75.0)	–	–	1	34.2 (23.9–45.7)	49.262	<0.001
Hepatobiliary cancer	3	64.3 (57.2–71.0)	–	–	–	–	–	–	–	–

* *p* < 0.1 (The null hypothesis is rejected; heterogeneity is present), ** k: Number of prevalence estimates, CIs: Confidence intervals, ^†^ The income level classifications (2021–2022) used in this study were obtained from the World Bank: high-income economies (>USD 12,695), upper-middle-income economies (USD 4096–USD 12,695), and lower-middle-income economies (USD 1046–USD 4095).

**Table 4 cancers-16-03130-t004:** Reason for non-disclosure and response of physician to disclosure of T&CM use.

Variables	No. of Studies Reporting Reasons	No. of Studies Indicating the Main Reason
**Reasons for Non-Disclosure of T&CM Use**	14 [46,48,50,52,58,59,62,63,68,69,70,73,79,81]	
1. Physicians did not ask about T&CM use	9 [46,48,52,59,63,68,69,70,73]	6 [46,48,52,63,68,69]
2. Patients did not feel the need to consult with physicians or consider it important to inform them	7 [46,50,62,63,68,70,79]	4 [50,62,70,79]
3. Physicians might discourage, disapprove, or object to T&CM use	9 [46,50,52,63,68,69,70,73,81]	-
4. Patients perceived T&CM as harmless	2 [73,81]	2 [73,81]
5. Patients may have lacked the time or chance to consult, or hesitated or forgot to disclose	6 [58,59,62,73,79,81]	1 [58]
6. Physicians may not understand why patients use T&CM	4 [46,59,62,79]	1 [59]
7. Patients’ anxiety that their doctor will be angry or their cancer treatment will be stopped	2 [50,59]	-
**Variables**	**No. of Studies Reporting** **Physician’s Response**	**The Pooled Estimate** **of Physician’s Response**
**Response of Physician to the Disclosure of T&CM Use**	7 [50,52,62,66,70,73,81]	
The physician encouraged the continuation of T&CM usage.	20.0 [66]–66.4% [73]	36.5% (95% CI = 21.7–52.5)
The physician neither endorsed nor discouraged the use of T&CM.	13.9 [50]–70.5% [54]	32.2% (95% CI = 18.2–47.9)
The physician advised against the use of T&CM.	4.9 [81]–65.0% [50]	26.0% (95% CI = 7.9–49.6)

## Data Availability

All datasets generated and analyzed, including the study protocol, search strategy, list of the included and excluded studies, data extracted, analysis plans, quality assessment, and assessment of the publication bias, are accessible by contacting the corresponding author of this study.

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
