# Peer review of "Traditional and Complementary Medicine Use among Cancer Patients in Asian Countries: A Systematic Review and Meta-Analysis"

_cancers, 2024, doi:10.3390/cancers16183130_

Round 1

Reviewer 1 Report

Comments and Suggestions for Authors

1. The title is rather misleading. The review is actually looking at studies from Asian countries and not strictly speaking done on only Asians since there is no way of ensuring the latter. Please revise the title accordingly.

2. "The primary contentious issues surrounding the use of traditional and complementary medicine (T&CM) by cancer patients, are noncompliance with treatment and potential hazards arising from Drug-T&CM interactions." - This is a convoluted and awkward sentence, suggest rephrasing please.

3. To provide a more balanced perspective in the introduction, it is relevant to include some comments about herbal therapies in general, which are gaining popularity amongst the public (citation: pubmed.ncbi.nlm.nih.gov/9820257). Some herbal therapies such as St John's wort and curcumin, also have well-demonstrated antidepressant effects (citation: pubmed.ncbi.nlm.nih.gov/28064110 and pubmed.ncbi.nlm.nih.gov/28236605) but caution has been advised regarding the potential for drug-drug interactions especially in the case of St John's wort.

4. There are significant discrepancies between what is described in the manuscript and the protocol deposited in PROSPERO. The protocol in PROSPERO focuses on "the prevalence and disclosure rate of Complementary Medicine use among patients with cancer in Asian countries" and uses a different set of databases and search terms. Any deviations from the original registration should be justified and the PROSPERO record should be updated in a timely fashion.

5. As per the PRISMA guidelines, please provide the full electronic search strategy used to identify studies, including all search terms and limits for at least one database in the main manuscript. The rest should be provided in the Supplementary Material.

6. The scoring system described for AXIS (a maximum score of 19 points) is not standard practice. AXIS has 20 items and typically acceptable reporting is taken to be ≥80% score on the AXIS tool.

7. The I² values for most analyses/forest plots were extremely high (e.g., 97%). This suggests substantial heterogeneity and also raises concerns about the appropriateness of the meta-analysis. Clearer descriptions of the sampling frame and outcome measurements are necessary. The extent of heterogeneity, more than just the pooled estimate, is vital to a meta-analysis of pooled prevalence. 

8. Although the authors conducted subgroup analyses and meta-regressions, the chosen subgroups (e.g., year of survey, sample size) requires further justification. More robust methods to explore and explain heterogeneity should be employed.

9. The discussion often implies causation (e.g., T&CM leading to non-compliance) without robust evidence. The language should be carefully revised to avoid misleading interpretations and sweeping statements.

10. The study's findings are generalized to all of Asia, which is an extremely diverse region with significant cultural, economic, and healthcare differences. This limitation should be acknowledged and discussed in greater detail.

Comments on the Quality of English Language

Moderate edits needed.

Author Response

Comments 1: The title is rather misleading. The review is actually looking at studies from Asian countries and not strictly speaking done on only Asians since there is no way of ensuring the latter. Please revise the title accordingly.

Response 1: Thank you for bringing this to our attention. We agree with your comments and have accordingly revised the title as follows: Traditional and complementary medicine use among cancer patients in Asian countries: A systematic review and meta-analysis.

Comments 2: "The primary contentious issues surrounding the use of traditional and complementary medicine (T&CM) by cancer patients, are noncompliance with treatment and potential hazards arising from Drug-T&CM interactions." - This is a convoluted and awkward sentence, suggest rephrasing please.

Response 2: We appreciate your feedback. We have revised the sentence for clarity as follows: "The primary concerns associated with the use of traditional and complementary medicine (T&CM) by cancer patients are noncompliance with conventional treatments and the potential risks of interactions between pharmaceuticals and T&CM."

Comments 3: To provide a more balanced perspective in the introduction, it is relevant to include some comments about herbal therapies in general, which are gaining popularity amongst the public (citation: pubmed.ncbi.nlm.nih.gov/9820257). Some herbal therapies such as St John's wort and curcumin, also have well-demonstrated antidepressant effects (citation: pubmed.ncbi.nlm.nih.gov/28064110 and pubmed.ncbi.nlm.nih.gov/28236605) but caution has been advised regarding the potential for drug-drug interactions especially in the case of St John's wort.

Response 3: Thank you for your valuable feedback. We have updated the introduction to provide a more balanced perspective by including comments on the popularity and therapeutic benefits of certain herbal therapies. The suggested change is made in line 65 – 70 as follows: Curcumin, commonly known as turmeric, exhibits anticancer activity across various cancer types [13]. Ginger is effective in alleviating chemotherapy-related nausea and vomiting [14], and St. John's wort is efficacious in managing depression and anxiety [15,16]. However, the use of T&CM requires caution due to potential drug interactions, which may attenuate therapeutic effects or exacerbate adverse drug reactions [16-18].

Comments 4: There are significant discrepancies between what is described in the manuscript and the protocol deposited in PROSPERO. The protocol in PROSPERO focuses on "the prevalence and disclosure rate of Complementary Medicine use among patients with cancer in Asian countries" and uses a different set of databases and search terms. Any deviations from the original registration should be justified and the PROSPERO record should be updated in a timely fashion.

Response 4: Thank you for pointing out the discrepancies between the manuscript and the PROSPERO protocol. We acknowledge the differences and have made the necessary updates to address them. The PROSPERO record has been revised to accurately reflect the changes and align with the manuscript. We have also updated our methods and search terms accordingly. Thank you for bringing this to our attention.

Comments 5: As per the PRISMA guidelines, please provide the full electronic search strategy used to identify studies, including all search terms and limits for at least one database in the main manuscript. The rest should be provided in the Supplementary Material.

Response 5: Thank you for bringing this to our attention. When we initially submitted the paper, Supplementary Table S1, "Search Strategies," was included as part of the Supplementary Files. If this file is missing or not accessible, please let us know. And also we will promptly resubmit the supplementary material to ensure it is available for review.

Comments 6: The scoring system described for AXIS (a maximum score of 19 points) is not standard practice. AXIS has 20 items and typically acceptable reporting is taken to be ≥80% score on the AXIS tool.

Response 6: Thank you for pointing this out. We agree with this comment regarding the scoring system of the AXIS tool. However, we determined that using a modified AXIS tool for critical appraisal does not limit the evaluation process.[1],[2] Therefore, we decided to remove the item on non-response bias, as it is not relevant to the studies included in our analysis. It should be noted that the interpretation of quality scores guided by the AXIS tool is subjective. In our analysis, we have referred to the guidelines in previous studies[3] to ensure consistency.

Comments 7: The I² values for most analyses/forest plots were extremely high (e.g., 97%). This suggests substantial heterogeneity and also raises concerns about the appropriateness of the meta-analysis. Clearer descriptions of the sampling frame and outcome measurements are necessary. The extent of heterogeneity, more than just the pooled estimate, is vital to a meta-analysis of pooled prevalence.

Response 7: The authors acknowledge the reviewer's concern regarding significant heterogeneity. To address this, we have provided additional details about the sampling frame and outcome measures in the Methods section. Specifically, we have revised line 106 –110 as follows: To mitigate population differences, a known source of heterogeneity, the sampling frame was restricted to cancer patients who were actively undergoing treatment at cancer centers or hospitals. Studies that included patients who were visiting for follow-up purposes after completing direct cancer treatments were excluded. Additionally, only studies in which the use of T&CM was explicitly reported as either "yes" or "no" were included in the analysis. Additionally, we have expanded the discussion of study heterogeneity in the Discussion section.

Comments 8: Although the authors conducted subgroup analyses and meta-regressions, the chosen subgroups (e.g., year of survey, sample size) requires further justification. More robust methods to explore and explain heterogeneity should be employed.

Response 8: We appreciate your feedback and agree with your observations. The authors have engaged in in-depth discussions to address these concerns. To assess trends in utilization, we conducted subgroup analyses by year. The survey years were categorized based on the publication of the WHO Report on Traditional Medicine Strategy 2014-2023, which highlights updates in health services and systems, including T&CM.

And to identify heterogeneity, subgroup analysis was conducted on various groups, including sample size. As noted in the quality assessment, many studies did not report calculating the appropriate sample size. Therefore, the authors attempted an analysis based on sample size to explore if there were differences depending on the sample size. However, in accordance with the reviewer's comments, the authors determined that subgroup comparisons based on sample size lacked meaning and thus excluded this analysis from the paper.

In response to your feedback, we have revised our approach. We conducted a new analysis focusing on variables identified in existing research as affecting estimates, such as data collection methods and whether sample size calculations were reported. These adjustments were incorporated into the paper to enhance the robustness of our estimation of T&CM use. Authors were made efforts to identify potential covariates contributing to observed heterogeneity and to stratify studies into more homogeneous subgroups, thereby improving the validity of our findings.

Comments 9: The discussion often implies causation (e.g., T&CM leading to non-compliance) without robust evidence. The language should be carefully revised to avoid misleading interpretations and sweeping statements.

Response 9: Thank you for pointing this out. We agree with this comment. Therefore, we revised discussion. 1) we deleted that paragraph “Findings from previous studies indicate that patients who received complementary therapies were more likely to refuse additional cancer treatment and face a higher risk of mortality [87], underscoring the importance of identifying features and factors associated with their usage.”

Comments 10: The study's findings are generalized to all of Asia, which is an extremely diverse region with significant cultural, economic, and healthcare differences. This limitation should be acknowledged and discussed in greater detail.

Response 10: We appreciate your observation regarding the diversity within Asia, which indeed impacts the generalizability of our study's findings. This aspect has been acknowledged and discussed in greater detail in the revised discussion section. Specifically, we have addressed this issue starting from line 436 in the revised manuscript. Thank you for highlighting this important consideration.

[1] McArthur, Katherine et al. “Epidemiology of Acute Injuries in Surfing: Type, Location, Mechanism, Severity, and Incidence: A Systematic Review.” Sports (Basel, Switzerland) vol. 8,2 25. 20 Feb. 2020, doi:10.3390/sports8020025

[2] Lundin, Rebecca et al. “Quality of routine health facility data used for newborn indicators in low- and middle-income countries: A systematic review.” Journal of Global Health vol. 12 04019. 23 Apr. 2022, doi:10.7189/jogh.12.04019

[3] Moor, L.; Anderson, J.R. A systematic literature review of the relationship between dark personality traits and antisocial online behaviours. Pers Individ Dif 2019, 144, 40-55.

Reviewer 2 Report

Comments and Suggestions for Authors

In the current review the authors determined the prevalence and disclosure rate of traditional and complementary medicine use among cancer patients in Asia. They concluded that the prevalence of traditional and complementary medicine use among cancer patients in Asian countries is high, but the disclosure rate of traditional and complementary medicine use to physicians is less.

Some suggestions:

1. In my opinion, a simple summary is not necessary in addition to the abstract. 2. pg 2, line 52: You give data from 2020. We are in 2024. Please update the information. 3. At materials and methods please add in what consists traditional and complementary medicine in Asian countries and which are the 14 Asian contries involved in the study.

4. Please give some details concerning the first exclusion criteria (point i), lines 110-111).

5. You forgot to add the Suppl Table S1

6. Who participated in sorting the articles? 7. The discussions are brief. 8. The results of the study are predictable. 9. The iThenticate report shows a 68% percent match. You must improve this aspect.  

Comments on the Quality of English Language

Minor editing of English language is required.

Author Response

In the current review the authors determined the prevalence and disclosure rate of traditional and complementary medicine use among cancer patients in Asia. They concluded that the prevalence of traditional and complementary medicine use among cancer patients in Asian countries is high, but the disclosure rate of traditional and complementary medicine use to physicians is less. Some suggestions:

Comments 1: In my opinion, a simple summary is not necessary in addition to the abstract.

Response 1: Thank you for comments. However, "Simple summary" is the one of section of the Cancers journal template. We agreed with your opinion and deleted it.

Comments 2: pg 2, line 52: You give data from 2020. We are in 2024. Please update the information.

Response 2: Thank you for pointing this out. We have updated the information accordingly. The revised sentence on page 2, line 49, now reads: “In 2022, an estimated 9.7 million individuals worldwide succumbed to cancer, with approximately 56.1% of these deaths occurring in Asia.” And we also updated the references to the latest.

Comments 3: At materials and methods please add in what consists traditional and complementary medicine in Asian countries and which are the 14 Asian countries involved in the study.

Response 3: Thank you for your suggestion. Due to the significant variation in the composition of traditional and complementary medicine across different countries and studies, we have outlined the overall composition in detail in Supplementary Table S5. Additionally, detailed information regarding the 14 Asian countries included in the study, as well as the traditional and complementary medicine practices within these countries, can be found in section 3.2. Study Characteristics.

Comments 4: Please give some details concerning the first exclusion criteria (point i), lines 110-111).

Response 4: Thank you for your comment. To mitigate population differences, a known source of heterogeneity, we restricted the sampling frame to cancer patients who were actively undergoing treatment at cancer centers or hospitals. Additionally, to allow for accurate comparisons of T&CM utilization and non-disclosure rates with international estimates, we excluded studies conducted in facilities that provided both conventional medicine and T&CM. For example, in the study by Broom, A et al. “The use of traditional, complementary and alternative medicine in Sri Lankan cancer care: results from a survey of 500 cancer patients,” the provision of Ayurvedic facilities was identified as a potential influencing factor, which is why such studies were excluded.

Comments 5: You forgot to add the Suppl Table S1

Response 5: Thank you for bringing this to our attention. When we initially submitted the paper, Supplementary Table S1 was included as part of the Supplementary Files. If this file is missing or not accessible, please let us know. And also we will promptly resubmit the supplementary material to ensure it is available for review.

Comments 6: Who participated in sorting the articles?

Response 6: Regarding the sorting of the articles, which is described in the “2.3. Selection Process section” (line 134) of our manuscript. We believe that the relevant information has been included. However, we would appreciate your guidance on whether any additional description or clarification is necessary.

Comments 7: The discussions are brief.

Response 7: We appreciate your feedback regarding the brevity of the discussion section. We agree that a more in-depth analysis is beneficial. In response, we have revised the discussion section with a focus on several key aspects. First, we have elaborated on the potential causes of heterogeneity in the studies and discussed the findings from the subgroup analyses that showed significant differences. In this process, additional variables were incorporated during data collection (face to face interview vs self-completed questionnaire), and the sample size variable was reanalyzed by categorizing studies based on whether they presented a sample size calculation. Additionally, we have included a qualitative assessment to identify unexplained sources of heterogeneity. Furthermore, in the limitations section, we have specifically addressed the need for future research on the reasons for non-disclosure of T&CM use.

Comments 8: The results of the study are predictable.

Response 8: We appreciate your feedback and agree that some of our findings align with existing research, which may render certain results predictable. However, our study contributes significantly by revealing specific characteristics and reasons for non-disclosure of T&CM use in Asian countries that were not previously identified in systematic reviews. Additionally, we have highlighted methodological limitations, providing valuable insights for future research. We believe that these contributions offer new perspectives and guidance for the field.

Comments 9: The iThenticate report shows a 68% percent match. You must improve this aspect. 

Response 9: We are very concerned and surprised by the 68% match reported by iThenticate. Prior to submission, we conducted a thorough check using a different tool provided by our institution, CopyKiller, which indicated a significantly lower similarity level. Additionally, the most recent version of our manuscript had a similarity rate of only 1%. Currently, we do not have access to iThenticate reports ourselves. To address this issue comprehensively, could you please provide details on the specific sections or content that contributed to this high similarity score? This information would greatly assist us in making necessary revisions and ensuring compliance with academic standards. Thank you for your understanding and assistance in this matter.

Reviewer 3 Report

Comments and Suggestions for Authors

Q1: Abstract: should be tailored to elaborate it.

Q2: The criteria are well-defined using the CoCoPop framework, which is appropriate for systemic reviews; information sources and search strategy: appears to be comprehensive, covering major database and including manual searches; selection process: involvement of multiple authors for screening and resolving disagreements through discussion ensures rigor and consistent; data collection and items: data extraction and the categories for descriptive data and clearly stated; risk of bias assessment: Using the AXIS tool for quality assessment is appropriate; statistical analysis: the use of the Metaprop command in Stata is suitable for meta-analyses of proportions. The use of Freeman-Turkey double arcsine transformation is appropriate to stabilize variances; heterogeneity: Apparently, there were a number of  heterogeneity among different studies. It seems that the authors have managed well with those heterogeneity by subgroup analysis. The authors also used funnel polts and Egger’s test to assess publication bias. Good Job!

Q3: Suggest to rewrite the introduction for redundancy, such as repeated statement about the Asia’s burden, weaken the readability of this paper.

Q4: please pay attention to some terms, such as descriptive data, data items, to make them all consistent.

Q5: CSS? Cross-sectional study?

Q6: Can extend your findings more in the discussion, and discuss more about the future research based on your findings. For example, studies on the reasons for non-disclosure of T&CM use.

Author Response

Comments 1: Abstract: should be tailored to elaborate it.

Response 1: Thank you for your feedback. We appreciate your suggestion to elaborate on the abstract. We have updated the abstract to provide a more detailed and comprehensive overview of the review.

Comments 2: The criteria are well-defined using the CoCoPop framework, which is appropriate for systemic reviews; information sources and search strategy: appears to be comprehensive, covering major database and including manual searches; selection process: involvement of multiple authors for screening and resolving disagreements through discussion ensures rigor and consistent; data collection and items: data extraction and the categories for descriptive data and clearly stated; risk of bias assessment: Using the AXIS tool for quality assessment is appropriate; statistical analysis: the use of the Metaprop command in Stata is suitable for meta-analyses of proportions. The use of Freeman-Turkey double arcsine transformation is appropriate to stabilize variances; heterogeneity: Apparently, there were a number of heterogeneity among different studies. It seems that the authors have managed well with those heterogeneities by subgroup analysis. The authors also used funnel plots and Egger’s test to assess publication bias. Good Job!

Response 2. We appreciate your encouraging comments and are glad that our study met your expectations.

Comments 3: Suggest to rewrite the introduction for redundancy, such as repeated statement about the Asia’s burden, weaken the readability of this paper.

Response 3: The suggested change is made in line 49–51 as follows: The growing impact of cancer is a pressing global healthcare concern with Asia bearing the highest burden. In 2022, an estimated 9.7 million individuals worldwide succumbed to cancer, with approximately 56.1% of these deaths occurring in Asia

Comments 4: please pay attention to some terms, such as descriptive data, data items, to make them all consistent.

Response 4: We agree with your suggestion and have unified the terminology for consistency. The suggested change is made in line 148 as follows: Data items were extracted across three categories:

Comments 5: CSS? Cross-sectional study?

Response 5: Yes, Abbreviations are described at the bottom of the table.

Comments 6: Can extend your findings more in the discussion, and discuss more about the future research based on your findings. For example, studies on the reasons for non-disclosure of T&CM use.

Response 6: Thank you for your valuable suggestion. We have revised the discussion section with a focus on several key aspects. First, we have elaborated on the potential causes of heterogeneity in the studies and discussed the findings from the subgroup analyses that showed significant differences. In this process, additional variables were incorporated during data collection (face to face interview vs self-completed questionnaire), and the sample size variable was reanalyzed by categorizing studies based on whether they presented a sample size calculation. Additionally, we have included a qualitative assessment to identify unexplained sources of heterogeneity. Furthermore, in the limitations section, we have specifically addressed the need for future research on the reasons for non-disclosure of T&CM use.

Round 2

Reviewer 1 Report

Comments and Suggestions for Authors

1. "... no attempts have been made to estimate" - Suggest to rewrite this as "... no prior review has tried to estimate".

2. The amendments to the original protocol and the reason(s) for these amendments should be detailed in the Methods section.

3. Please change "a confidence interval of 95% (CI 95%)" to "a confidence interval of 95% (95% CI)."

4. At least some justification should be provided for the specific subgroups (gender, geographic location, and insurance coverage) chosen for analysis.

5. The authors' conclusions are not clearly supported by the data presented. For example, the suggestion that integrating T&CM into conventional cancer care should be a priority is not sufficiently justified by the findings as your study does not speak about the synergistic effectiveness of T&CM with Western medicine. Please adjust accordingly.

Comments on the Quality of English Language

Moderate changes required.

Author Response

Comments 1: "... no attempts have been made to estimate" - Suggest to rewrite this as "... no prior review has tried to estimate".

Response 1: We appreciate your suggestions and have made the recommended changes accordingly. We have revised the phrase from "... no attempts have been made to estimate" to "... no prior review has tried to estimate" to improve clarity and accuracy.

Comments 2: The amendments to the original protocol and the reason(s) for these amendments should be detailed in the Methods section.

Response 2: Thank you for your insightful comments. When the protocol was initially registered, the title included "Complementary medicine use." However, after referring to WHO guidelines and through internal discussions and advice, we recognized that in Asia, the terms "complementary medicine" and "traditional medicine" are often used interchangeably. As a result, we changed the title to "Traditional and complementary medicine use among cancer patients in Asian countries: A systematic review and meta-analysis." Additionally, the publication search period was extended to 2023 to include the most recent studies. Since there were no other significant changes in the search methods or analysis, we did not detail these adjustments separately in the Methods section. We believe that including such details would be unnecessary and may not fully align with the reviewer's intent.

Comments 3: Please change "a confidence interval of 95% (CI 95%)" to "a confidence interval of 95% (95% CI)."

Response 3: Thank you for your valuable feedback. We have updated the terminology to "a confidence interval of 95% (95% CI)" as suggested.

Comments 4: At least some justification should be provided for the specific subgroups (gender, geographic location, and insurance coverage) chosen for analysis.

Response 4:  Thank you for your insightful feedback regarding the justification for the specific subgroups chosen for analysis. 
1. The suggested change has been made in lines 192-194: Given Asia's extensive diversity, including variations in cultural, economic, and healthcare contexts, the region was classified into four distinct geographic areas to evaluate potential differences in T&CM use [8] 
2. The suggested change has been made in lines 472-475: Health insurance coverage of T&CM is an emerging topic that reflects trends in T&CM utilization and serves as a significant factor influencing the high prevalence estimates of these practices among cancer patients, while also allowing for national-level comparison of information.
3. The suggested change has been made in lines 487-489: Gender is one of the most common demographic predictors of T&CM use among cancer patients [28] 

Comments 5:  The authors' conclusions are not clearly supported by the data presented. For example, the suggestion that integrating T&CM into conventional cancer care should be a priority is not sufficiently justified by the findings as your study does not speak about the synergistic effectiveness of T&CM with Western medicine. Please adjust accordingly.

Response 5: Thank you for your valuable feedback regarding the conclusions drawn from our study. We have revised the conclusions to ensure they are adequately supported by the data presented.  The suggested change has been made in lines 605-614: T&CM is widely used among cancer patients in Asia, with usage associated with demographic factors, geographical regions, and health insurance coverage. However, many patients do not disclosure their T&CM use to healthcare providers, often due to in-sufficient physician inquiry and a perceived lack of necessity.    This underscores the critical need to improve patient-physician communication to ensure effective cancer treatment and patient well-being. Enhanced communication can improve medication ad-herence and pain management, thereby promoting patient-centered care. Future research should identify barriers to T&CM disclosure and their impact on treatment adherence, supporting a more integrative and patient-centered cancer care approach.

We have also made further revisions to the English expressions. We are pleased to hear that you find the manuscript suitable for publication, and we are grateful for your support.

Thank you once again for your time and valuable input.

Reviewer 2 Report

Comments and Suggestions for Authors

The manuscript has been significantly improved. The authors addressed all the required issues and in my opinion, the manuscript is suitable for publication.

Comments on the Quality of English Language

Minor editing of English language is required.

Author Response

Comments 1: The manuscript has been significantly improved. The authors addressed all the required issues and in my opinion, the manuscript is suitable for publication.

Response 1: Thank you very much for your positive feedback and for acknowledging the improvements made to the manuscript. We greatly appreciate your thoughtful comments and suggestions throughout the review process, which have significantly contributed to enhancing the quality of our work. We have also made further revisions to the English expressions. We are pleased to hear that you find the manuscript suitable for publication, and we are grateful for your support.

Thank you once again for your time and valuable input.

Round 3

Reviewer 1 Report

Comments and Suggestions for Authors

Thank you for the replies and edits.

Comments on the Quality of English Language

Slight proofreading required.